# Cerebellar Purkinje cell stripe patterns reveal a differential vulnerability and resistance to cell loss during normal aging in mice

Sarah G Donofrio[1,2,3,4], Cheryl Brandenburg[1,3,4], Amanda M Brown[1,3,4], Tao Lin[1,3,4], Hsiang-Chih Lu[1,3,4], Roy V Sillitoe[1,2,3,4,5]*

[1]Department of Pathology and Immunology, Baylor College of Medicine, Houston, United States; [2]Department of Neuroscience, Baylor College of Medicine, Houston, United States; [3]Cerebellum Science Center, Texas Children's Hospital, Houston, United States; [4]Jan and Dan Duncan Neurological Research Institute, Texas Children's Hospital, Houston, United States; [5]Department of Pediatrics, Baylor College of Medicine, Houston, United States

## eLife Assessment

This **important** study presents findings on the patterned loss of Purkinje cells in the cerebellum during aging. The **compelling** data nicely support the conclusions of this study. This work advances understanding of mechanisms underlying neurodegeneration with aging and provides the basis for development of treatments for age-related neurological disorders.

*For correspondence:
sillitoe@bcm.edu

**Abstract** Age-related neurodegenerative diseases involve reduced cell numbers and impaired behavioral capacity. Neurodegeneration and behavioral deficits also occur during aging, and notably in the absence of disease. The cerebellum, which modulates movement and cognition, is susceptible to cell loss in both aging and disease. Here, we demonstrate that cerebellar Purkinje cell loss in aged mice is not spatially random but rather occurs in a pattern of parasagittal stripes. We also find that aged mice exhibit impaired motor coordination and more severe tremor compared to younger mice. However, the relationship between patterned Purkinje cell loss and motor dysfunction is not straightforward. Examination of postmortem samples of human cerebella from neurologically typical individuals supports the presence of selective loss of Purkinje cells during aging. These data reveal a spatiotemporal cellular substrate for aging in the cerebellum that may inform how neuronal vulnerability leads to neurodegeneration and the ensuing deterioration of behavior.

## Introduction

In addition to its well-known roles in motor function, the cerebellum is also involved in cognitive functions that include executive function, visuospatial memory, language, and emotional processing (*Stoodley et al., 2012*; *Strick et al., 2009*; *Reeber et al., 2013*; *Rudolph et al., 2023*; *Timmann and Daum, 2007*; *Berlijn et al., 2024*; *Schmahmann, 2019*; *Kim et al., 2024*; *Clausi et al., 2022*; *Adamaszek et al., 2017*; *Van Overwalle et al., 2020*). Accordingly, the cerebellum is a major culprit in movement disorders (*Marsden, 2018*; *Klockgether et al., 2019*; *Holmes, 1917*; *Holmes, 1939*; *Marin-Lahoz and Gironell, 2016*; *Louis, 2016*; *Benito-León and Labiano-Fontcuberta, 2016*;

**eLife digest** Aging involves the gradual loss of brain cells and a decline in physical and cognitive capabilities. One brain region affected by aging is the cerebellum, which is best known for its role in motor coordination.

During healthy aging, the cerebellum is reduced in size, and elderly individuals often face difficulties with balance and motor skills. This shrinkage is partly caused by the loss of neurons, particularly Purkinje cells. A reduction in Purkinje cells and impaired motor function have also been seen in rodents during normal aging.

Furthermore, in many rodent models of disease, not all Purkinje cells are equally likely to die. The surviving cells form specific patterns similar to those seen during cerebellar development. However, it has so far been unclear whether such regional differences in Purkinje cell loss also occur during healthy aging. Addressing this question would shed light on how aging influences cellular susceptibility and resilience in the cerebellum and how these cellular responses affect motor function and age-related cell death.

To find out whether loss of Purkinje cells in aged mice occurs in a similar pattern observed during disease, Donofrio et al. used a combination of genetic, histological, and imaging techniques to visualize Purkinje cells in the cerebellum.

The results revealed that the loss of Purkinje cells in healthy aged mice occurred in a pattern similar to that often observed in models of disease. However, the overall pattern in aged mice was distinct and occurred in a parasagittal, striped pattern – that is, long, narrow, and running front-to-back through the cerebellum. In addition, examining human cerebellum tissue samples collected from individuals without any reported neurological or neuropsychiatric problems confirmed a loss of Purkinje cells that increased with age. However, a specific pattern remains to be confirmed.

Our study reveals a cerebellar framework of vulnerability and resistance to age-related cell death. These findings could enhance healthy brain aging by improving the precision of targeted therapeutics and opening avenues for preventative strategies to reduce or prevent cell loss. The essential step is to fully understand when and how cerebellar neurons degenerate across the human lifespan.

*Handforth, 2016*; *Filip et al., 2016*; *Cerasa and Quattrone, 2016*; *Shakkottai et al., 2017*; *Louis and Faust, 2020*; *Mukherjee and Pandey, 2024*; *Merola et al., 2019*; *Kumar et al., 2023*; *van der Heijden and Sillitoe, 2023*), and it likely also contributes to autism spectrum disorders (*Sydnor and Aldinger, 2022*; *Kelly et al., 2021*; *Su et al., 2021*; *Hampson and Blatt, 2015*; *D'Mello and Stoodley, 2015*; *Rogers et al., 2013*; *Fatemi et al., 2012*), sleep disturbances (*Jackson and Xu, 2023*; *Torres-Herraez et al., 2022*; *Xu et al., 2021*; *Salazar Leon et al., 2024*; *Salazar Leon and Sillitoe, 2022*), and schizophrenia (*Faris et al., 2024*; *Khalil et al., 2022*; *Cao and Cannon, 2019*). Despite the prevalence of cerebellar involvement in disease, the cerebellum is often overlooked in the context of aging. This omission has not gone unnoticed, with recent arguments for incorporating the cerebellum into our understanding of brain aging, which has historically focused on the cerebral cortex and hippocampus (*Bernard, 2022*). In support of this hypothesis, motor and cognitive behaviors are often impaired during normal aging. The decline in these behaviors could, in theory, be accompanied by cerebellar pathology, such as variations in cerebellar volume and alterations in its circuit connectivity (*Bernard and Seidler, 2014*; *Arleo et al., 2024*; *Iskusnykh et al., 2024*). Therefore, the cerebellum is a potentially critical model system for better understanding the structure and function of the brain during aging.

Although neurodegeneration is associated with disease pathogenesis, cerebellar degeneration can also occur during aging in the absence of disease. Elderly patients have decreased cerebellar volume compared to young patients (*Raz et al., 1998*; *Raz et al., 2001*; *Raz et al., 2013*; *Jernigan et al., 2001*), and longitudinal studies have shown decreased cerebellar volume in healthy older individuals over time (*Tang et al., 2001*; *Raz et al., 2005*; *Smith et al., 2007*; *Raz et al., 2010*; *Han et al., 2020*). At the cellular level, otherwise healthy aged patients have significantly decreased Purkinje cell density (*Andersen et al., 2003*; *Torvik et al., 1986*; *Sjobeck et al., 1999*). Depending on severity, age-related cerebellar atrophy can have detrimental and often debilitating effects on motor function. For example, decreased cerebellar volume in the elderly is correlated with impaired eyeblink

conditioning, a cerebellum-dependent associative learning task, as well as slower gait and impaired balance, two aspects of motor function that involve the cerebellum (*Woodruff Pak et al., 2001*, *Rosano et al., 2007*). Furthermore, there are reported structure-function relationships between the volume of specific cerebellar regions and the performance of sensorimotor tasks in young and old participants (*Bernard and Seidler, 2013*). These results suggest that cerebellar atrophy is associated with deficits in cerebellum-related functions during normal aging. However, the cellular nature of this structure-function relationship is still poorly understood.

Similar to humans, control mice and rats also experience Purkinje cell loss during aging (*Bakalian et al., 1991*; *Hadj-Sahraoui et al., 1996*; *Hadj-Sahraoui et al., 1997*; *Doulazmi et al., 1999*; *Doulazmi et al., 2006*; *Sturrock, 1989a*; *Sturrock, 1989b*; *Sturrock, 1990*; *Zanjani et al., 2004*; *Childs et al., 2021*), as well as motor dysfunction. At around 12 months of age, mice begin to display impaired performance on the rotarod task, a test of motor coordination and motor learning (*Caston et al., 1995*; *Vogel et al., 2002*; *Shoji and Miyakawa, 2019*), as well as impaired eyeblink conditioning (*Vogel et al., 2002*), now a classical test for Pavlovian learning. Interestingly, there is evidence to suggest that mice have significantly decreased Purkinje cell numbers starting at 18 months, corresponding with impaired delay eyeblink conditioning (*Woodruff Pak, 2006*). These studies suggest that cerebellum-associated motor function declines with age in mice and that this decline may be accompanied by Purkinje cell loss. However, the precise regionality of the cerebellum was not considered in these studies, and we argue that age-related changes follow a fundamental scheme.

Compared to the patterned Purkinje cell loss reported in different disease models, relatively little is known about how Purkinje cell loss affects different regions of the cerebellum during normal aging. Neuroimaging in humans has revealed that cerebellar volume in different lobules can be differentially affected by aging (*Raz et al., 1998*; *Han et al., 2020*; *Bernard and Seidler, 2013*; *Luft et al., 1999*; *Hulst et al., 2015*; *Yu et al., 2017*; *Wang et al., 2024*). However, a closer examination of any region-specific cellular differences is lacking. Other studies have found minimal regional differences beyond the observation that most age-related Purkinje cell loss seems to occur in the anterior cerebellum (*Andersen et al., 2003*, *Torvik et al., 1986*). In fact, multiple studies have found that Purkinje cell loss is uniform across the latero-lateral extent of the cerebellar cortex in rodents (*Bakalian et al., 1991*; *Hadj-Sahraoui et al., 1996*; *Hadj-Sahraoui et al., 1997*; *Doulazmi et al., 1999*; *Doulazmi et al., 2006*; *Sturrock, 1990*). However, cerebellar organization and patterning are much more complex than the broad medio-lateral and anterior-posterior differences can account for.

The cerebellum is highly compartmentalized on multiple levels as determined by genetics and developmental, anatomical, and electrophysiological studies. Based on development and gene expression boundaries, the cerebellum can be divided into transverse zones: the anterior (lobules I-V), central (lobules VI and VII), posterior (lobule VIII and anterior lobule IX), and nodular (posterior lobule IX and lobule X) zones (*Ozol et al., 1999*). Within each transverse zone, subpopulations of Purkinje cells are divided into stripes based on gene expression patterns, which differ between transverse zones (*Marzban et al., 2004*; *Sarna et al., 2006*; *Chung et al., 2008*; *Demilly et al., 2011*; *Brochu et al., 1990*; *Armstrong et al., 2000*; *Dehnes et al., 1998*). Numerous stripe markers exist, with overlapping, partially overlapping, or unique expression patterns. The most well-studied stripe marker is zebrin II (*Hawkes and Herrup, 1995*; *Ebner et al., 2012*; *Gravel et al., 1987*), the expression pattern of which is remarkably consistent from animal to animal and conserved across mammals (*Sillitoe et al., 2005*; *Sillitoe and Joyner, 2007*). The identity of individual Purkinje cells, which can express different combinations of patterned markers, is established when each Purkinje cell is born (*Hashimoto and Mikoshiba, 2003*; *Namba et al., 2011*; *Dastjerdi et al., 2012*; *Tran-Anh et al., 2020*), although the exact relationship between developmental Purkinje cell clusters and adult stripes has been only partially resolved (*Larouche and Hawkes, 2006*). The cerebellar stripe patterns represent one component of a greater network architecture. For example, longitudinal groups of Purkinje cells are related in the rostrocaudal axis based on specific inputs from the inferior olive through climbing fibers and specific outputs to the cerebellar nuclei through the Purkinje cell axons (*Apps and Hawkes, 2009*). Each longitudinal zone, together with its efferent and afferent pathways, comprises a functional unit referred to as a cerebellar module (*Ruigrok, 2011*). Based on these overlapping levels of organization, the cerebellum is predicted to be composed of hundreds or thousands of modules. This hypothesis posits that the multiple maps of cerebellar compartmentation that are visualized with different approaches comprise a single overarching map, with alignment between zones, stripes, and

modules (*Apps and Hawkes, 2009*). Here, we wondered whether the patterns that are revealed by stripe markers reflect a map of Purkinje cell degeneration and eventual cell loss with age and if there is a functional significance to this patterned cellular demise.

To test for and investigate the potential contribution of patterned Purkinje cell loss to aging in mice, we used a combination of whole-mount immunohistochemistry (*Sillitoe and Hawkes, 2002*; *White et al., 2012*) and analysis of Purkinje cell patterns on histological tissue slices. These techniques enabled us to visualize Purkinje cells across the surfaces of entire cerebella of normal aged mice and examine their cellular level changes, respectively. We also used genetically driven fluorescent reporter labeling to specifically label Purkinje cells. Using these techniques, we uncovered that some, but not all, aged mice have Purkinje cell loss that occurs in a pattern of parasagittal stripes. In addition, upon immunostaining coronal sections of cerebellar tissue, we found that the pattern of age-related Purkinje cell loss is unique compared to the most common patterns of Purkinje cell loss that have been reported in numerous mouse models of disease. Behavioral tests revealed deficits in motor function in aged mice compared to young mice. Finally, we observed patches of Purkinje cell degeneration in postmortem tissue obtained from neurologically normal humans. We therefore have three key findings to report in this series of studies: (1) Purkinje cell degeneration and cell loss are prominent features of the otherwise healthy aging mouse cerebellum, (2) Purkinje cell loss during normal aging in mice does not occur in a random manner and instead occurs in an array of parasagittal stripes that reflect the normal developmental, anatomical, and functional topography of the mammalian cerebellum, and (3) this mode of patterned cell loss may be conserved and reflect a fundamental heterogeneity of the human cerebellum that distinguishes cells that are vulnerable versus resistant to deterioration in health and disease. Together, these data underscore the potential clinical significance of our findings of patterned Purkinje cell degeneration and loss in normal aging mice, as they may inform the design and development of effective therapies for specific neurological and neuropsychiatric conditions that are defined by compromised cerebellar structure and function.

## Results

Adult-onset neurodegeneration typically impacts specific regions of the brain. For instance, in diseases such as Alzheimer's disease and Parkinson's disease, neurodegeneration affects the cortex and basal ganglia, resulting in defective cognitive versus motor circuits, respectively. A similar phenomenon occurs during normal aging, as certain brain regions are more susceptible to degeneration (*Pandya and Patani, 2021*). Interestingly, the cerebellum is a target in age-related degeneration, and its cellular demise can lead to both cognitive and motor impairments. We examine whether different populations of cells are more susceptible than others. What dictates this regional vulnerability and whether these processes and sensitive neuronal subpopulations are the same across different diseases and typical aging is unknown. To begin to address this problem, here we sought to test whether cerebellar Purkinje cell loss follows a region-specific pattern during normal aging.

### Aged mice have Purkinje cell loss that occurs in parasagittal stripes

We started by asking whether normal mice exhibit patterned Purkinje cell loss during aging. Previous studies have relied on tissue sections alone for identifying the presence of degeneration and cell loss, although studying the complexity of the cerebellum on individual slices limits one's ability to visualize topographic patterns. In our study, to test whether age-related Purkinje cell loss is indeed patterned, we examined a total of 52 cerebella from aged mice (between 13 and 25 months of age; *Table 1*). We used a combination of techniques to visualize Purkinje cells, including whole-mount immunohistochemistry (*Sillitoe and Hawkes, 2002*; *White et al., 2012*) with calbindin antibodies (n=5), whole-mount and light sheet imaging of a Purkinje-cell-specific reporter in transgenic mice (n=21 and n=2, respectively), and histology on coronal tissue sections (n=34; *Table 1*). Using these techniques, we observed that Purkinje cell loss across the cerebella of aged mice is not uniform, as has been previously reported (*Bakalian et al., 1991*; *Hadj-Sahraoui et al., 1996*; *Hadj-Sahraoui et al., 1997*; *Doulazmi et al., 1999*; *Doulazmi et al., 2006*; *Sturrock, 1990*), but forms a pattern. Importantly, the pattern is composed of parasagittal stripes that are symmetrical about the midline. In the calbindin-labeled whole-mount cerebella, the stripes appear as alternating dark stripes of surviving Purkinje cells and light stripes where Purkinje cells have presumably degenerated (*Figure 1A*). The

**Table 1.** Information from mice used in this study.
Mice from the same litter are indicated with the same letter.

| Sex | Age (months) | PC loss | Genotype | Littermates | Technique |
|---|---|---|---|---|---|
| M | 7 | striped PC loss | Pcp2$^{Cre+/-}$;ROSA26$^{Ai40D}$ | a | GFP whole-mount |
| F | 7.2 | no PC loss | Pcp2$^{Cre+/-}$;ROSA26$^{Ai40D}$ | | GFP whole-mount |
| M | 7.4 | no PC loss | no mutant alleles | a | coronal sections |
| F | 11.7 | striped PC loss | Pcp2$^{Cre+/-}$;ROSA26$^{Ai40D}$ | b | GFP whole-mount |
| M | 13.7 | no PC loss | ROSA26$^{Ai32+/+}$ | | coronal sections |
| M | 13.9 | striped PC loss | no mutant alleles | c | coronal sections |
| M | 14.3 | non-striped PC loss | Pcp2$^{Cre+/-}$;ROSA26$^{Ai32}$ | | GFP whole-mount |
| M | 14.3 | striped PC loss | Pcp2$^{Cre+/-}$;ROSA26$^{Ai32+/+}$ | d | GFP whole-mount |
| M | 14.4 | no PC loss | Pcp2$^{Cre+/-}$;ROSA26$^{Ai3\ +/+}$ | d | GFP whole-mount |
| F | 14.5 | striped PC loss | Pcp2$^{Cre+/-}$;ROSA26$^{Ai32+/-}$ | | GFP whole-mount |
| M | 14.7 | striped PC loss | Tau$^{lsl-mGFP-lacZ}$ | | calbindin whole-mount |
| F | 15.1 | striped PC loss | Pcp2$^{Cre+/-}$;ROSA26$^{Ai32}$ | | GFP whole-mount |
| M | 15.3 | striped PC loss | Pcp2$^{Cre+/-}$;Tau$^{lsl-mGFP-lacZ}$ | e | calbindin whole-mount |
| F | 15.3 | no PC loss | Tau$^{lsl-mGFP-lacZ}$;ROSA2$^{lsl-DTR+/-}$ | e | calbindin whole-mount |
| M | 15.7 | non-striped PC loss | Pdx1$^{Cre+/-}$ | | coronal sections |
| F | 16.3 | striped PC loss | Tau$^{lsl-mGFP-lacZ}$;ROSA26$^{lsl-DTR+/-}$ | | calbindin whole-mount |
| F | 16.4 | no PC loss | Tau$^{lsl-mGFP-lacZ}$ | | coronal sections |
| F | 16.4 | striped PC loss | Pcp2$^{Cre+/-}$;ROSA26$^{Ai40D+/-}$ | f | GFP whole-mount, coronal sections |
| M | 16.5 | non-striped PC loss | ROSA26$^{Ai32+/-}$ | g | coronal sections |
| M | 16.5 | no PC loss | Gdnf$^{CreER+/-}$;ROSA26$^{Ai32+/-}$ | g | coronal sections |
| M | 16.5 | non-striped PC loss | ROSA26$^{Ai32+/-}$ | g | coronal sections |
| M | 16.6 | no PC loss | Tau$^{lsl-mGFP-lacZ}$;ROSA26$^{lsl-DTR+/-}$ | | coronal sections |
| F | 16.8 | striped PC loss | Pcp2$^{Cre+/-}$;ROSA26$^{Ai40D}$ | | GFP whole-mount, coronal sections |
| M | 16.8 | no PC loss | no mutant alleles | h | coronal sections |
| M | 16.8 | no PC loss | no mutant alleles | h | coronal sections |
| M | 16.9 | no PC loss | Pcp2$^{Cre+/-}$ | i | coronal sections |
| M | 16.9 | no PC loss | Tau$^{lsl-mGFP-lacZ}$;ROSA26$^{lsl-DTR+/-}$ | i | coronal sections |
| M | 16.9 | no PC loss | no mutant alleles | j | coronal sections |
| M | 16.9 | no PC loss | no mutant alleles | j | coronal sections |
| M | 17 | striped PC loss | Pcp2$^{Cre+/-}$;ROSA26$^{Ai40D+/-}$ | k | GFP whole-mount, light sheet |
| M | 17.2 | striped PC loss | ROSA26$^{lsl-DTR+/-}$ | c | coronal sections |
| M | 17.3 | no PC loss | ROSA26$^{Ai40D+/-}$ | k | GFP whole-mount, coronal sections |
| M | 17.4 | striped PC loss | Pcp2$^{Cre+/-}$;ROSA26$^{Ai40D+/-}$ | k | GFP whole-mount |
| M | 17.4 | no PC loss | Ai32$^{+/+}$;VGAT$^{+/fx}$ | | coronal sections |
| F | 18 | striped PC loss | Pcp2$^{Cre+/-}$;ROSA26$^{Ai32}$ | b | GFP whole-mount |
| M | 18.3 | no PC loss | Mash1$^{CreER+/-}$;Tau$^{lsl-mGFP-lacZ}$;VGAT$^{fx/fx}$ | | coronal sections |
| F | 18.3 | striped PC loss | Pcp2$^{Cre+/-}$;ROSA26$^{Ai40D+/-}$ | l | GFP whole-mount |
| F | 18.5 | striped PC loss | Pcp2$^{Cre+/-}$;ROSA26$^{Ai32}$ | m | coronal sections |

*Table 1 continued on next page*

*Table 1 continued*

| Sex | Age (months) | PC loss | Genotype | Littermates | Technique |
|---|---|---|---|---|---|
| F | 18.9 | no PC loss | *ROSA26*$^{Ai40D+/-}$ | l | coronal sections |
| F | 19.8 | non-striped PC loss | *Tau*$^{lsl-mGFP-lacZ}$;*VGAT*$^{fx/fx}$ | | coronal sections |
| F | 20.1 | non-striped PC loss | *Tau*$^{lsl-mGFP-lacZ}$;*VGAT*$^{fx/fx}$ | n | coronal sections |
| F | 20.1 | non-striped PC loss | *Tau*$^{lsl-mGFP-lacZ}$;*VGAT*$^{fx/fx}$ | n | coronal sections |
| F | 20.9 | striped PC loss | *Pcp2*$^{Cre+/-}$;*ROSA26*$^{Ai32}$ | m | coronal sections |
| F | 20.9 | striped PC loss | *Pcp2*$^{Cre+/-}$;*ROSA26*$^{Ai32}$ | m | coronal sections |
| M | 20.9 | striped PC loss | *Pcp2*$^{Cre+/-}$;*ROSA26*$^{Ai32}$ | m | coronal sections |
| M | 21.2 | striped PC loss | *Pcp2*$^{Cre+/-}$;*ROSA26*$^{Ai32}$ | b | GFP whole-mount, coronal sections |
| M | 21.2 | striped PC loss | *Pcp2*$^{Cre+/-}$;*ROSA26*$^{Ai32}$ | b | GFP whole-mount, coronal sections |
| M | 21.4 | striped PC loss | *Pcp2*$^{Cre+/-}$;*ROSA26*$^{Ai40D}$ | o | GFP whole-mount, coronal sections |
| M | 21.4 | striped PC loss | *Pcp2*$^{Cre+/-}$;*ROSA26*$^{Ai40D}$ | o | GFP whole-mount, light sheet |
| F | 23.3 | no PC loss | *VGAT*$^{fx/fx}$ | | calbindin whole-mount |
| F | 25.5 | striped PC loss | *Pcp2*$^{Cre+/-}$;*ROSA26*$^{Ai40D+/-}$ | f | GFP whole-mount, coronal sections |
| M | 25.5 | striped PC loss | *Pcp2*$^{Cre+/-}$;*ROSA26*$^{Ai40D}$ | f | GFP whole-mount, coronal sections |

stripes are visible in the anterior, central, and posterior zones of the cerebellum, as well as in the paraflocculi. Closer inspection of the whole-mount cerebella revealed the shapes of individual Purkinje cells within the dark stripes, with the dendrites and cell bodies clearly visible (*Figure 1B*). Purkinje cell axons with torpedoes, a pathological sign of Purkinje cell neurodegeneration in disease (*Louis et al., 2009*) and normal aging (*Bäurle and Grüsser-Cornehls, 1994*), were also observed in whole-mount cerebella (*Figure 1B*). The presence of the axonal pathology suggests that the observed Purkinje cell loss in aged mice may be due to and potentially accompanied by a process of neurodegeneration that causes cell loss over time.

We observed considerable variability in terms of Purkinje cell loss in the aged mice. Some aged mice displayed a lack of Purkinje cell loss comparable to young control mice, even at 23 months of age (*Figure 1D–F*). Of the 52 aged cerebella examined, 19 lacked appreciable Purkinje cell loss, 26 had clearly striped Purkinje cell loss, and 7 had Purkinje cell loss that did not appear striped (*Table 1*; *Figure 1—figure supplement 1A*). This inconsistency in the loss of Purkinje cells is not due solely to relative age, as a 14-month-old cerebellum had striped Purkinje cell loss (*Figure 1G*), whereas a 23-month-old cerebellum did not (*Figure 1F*; *Table 1*). Despite this, when striped Purkinje cell loss did occur in the aged mice, the neurodegeneration appears progressive, meaning that older mice were more likely to display more widespread Purkinje cell loss compared to younger mice (*Figure 1G–I*). Aged mice with striped Purkinje cell loss consistently displayed the same pattern of neurodegeneration (*Figure 1G–I*). Therefore, Purkinje cells that are more susceptible to age-related neurodegeneration may belong to the same subpopulation of neurons and have the same identity across different mice.

Of the 52 aged mice whose cerebella were examined, 32 were male and 20 were female. We found that 12 of the 20 aged female mice and 14 of the 32 aged male mice had striped Purkinje cell loss (*Table 1*; *Figure 1—figure supplement 1A*). Analysis of the data with Fisher's exact test revealed that the presence of striped Purkinje cell loss is not sex dependent. However, a greater percentage of the aged females displayed Purkinje cell loss compared to the aged males (*Figure 1—figure supplement 1A*), suggesting that females may be more susceptible to Purkinje cell loss during aging. Overall, these results suggest that there is considerable variability plus context specificity in the spatiotemporal features of Purkinje cell degeneration, eventual Purkinje cell loss, and their associated pattern among aged mice.

Interestingly, among mice born in the same litter, we found that one littermate could have striped Purkinje cell loss while the other littermate did not (*Figure 1E and H*). This observation was especially

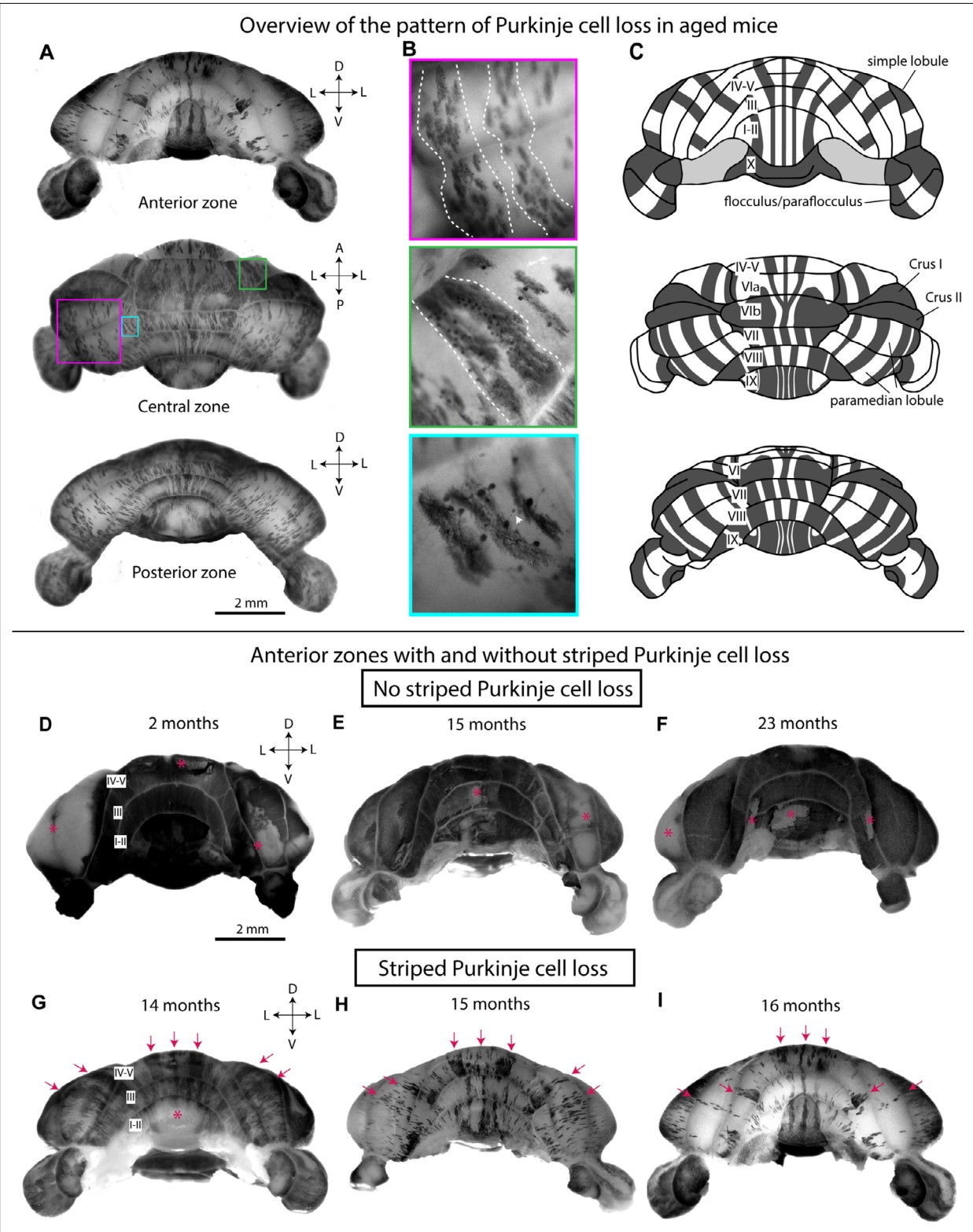

**Figure 1.** Whole-mount immunohistochemistry of the cerebellum reveals striped Purkinje cell loss across the cerebellar cortex of some aged mice. (**A**) Whole-mount cerebellum of a 16-month-old mouse immunostained for calbindin and viewed from different angles. D=dorsal; L=lateral; V=ventral; A=anterior; P=posterior. Scale bar = 2 mm. (**B**) High-magnification images of Purkinje cells in the whole-mount cerebellum of an aged mouse. Dotted lines indicate stripes of surviving Purkinje cells, and the white arrowhead indicates an axonal torpedo. (**C**) Schematic of the pattern of age-related Purkinje cell loss based on whole-mount cerebella, where dark gray stripes represent bands largely composed of surviving Purkinje cells and

*Figure 1 continued on next page*

*Figure 1 continued*

white stripes represent bands where most Purkinje cells have degenerated. Cerebellar lobules are labeled with Roman numerals. (**D–I**) Wholemount cerebella of mice immuno-stained for calbindin and viewed from the anterior zone: (**D**) 2-month-old mouse; (**E**) 15-month-old mouse without Purkinje cell loss; (**F**) 23-month-old mouse without Purkinje cell loss; (**G**) 14-month-old mouse with striped Purkinje cell loss; (**H**) 15-month-old mouse with striped Purkinje cell loss; (**I**) 16-month-old mouse with striped Purkinje cell loss. Scale bar = 2 mm. Asterisks indicate staining artifacts, either caused by continuous rubbing of the cerebellum during staining (hemispheres in panels D-F) and lobules I-II in panel (**G**) or accidental removal of surface tissue during dissection (lobules I-II in panel F). Arrows indicate bands of surviving Purkinje cells that are consistent across mice with striped Purkinje cell loss. D=dorsal; L=lateral; V=ventral. Cerebellar lobules are labeled with Roman numerals.

The online version of this article includes the following source data and figure supplement(s) for figure 1:

**Figure supplement 1.** The presence of Purkinje cell loss varies across aged mice.

**Figure supplement 1—source data 1.** Source data for *Figure 1—figure supplement 1*.

evident as observed on surface mapping using the whole-mount immunohistochemical staining approach (*Figure 1*). This difference in cell loss was observed in five sets of littermates, with each set from a different litter (*Table 1*). These data suggest that even genetically similar mice raised in the same cage can display dramatic differences in age-related neurodegeneration.

Taken together, these findings suggest that (1) Purkinje cell subpopulations are differentially vulnerable to death during normal aging; (2) the loss of Purkinje cells, according to age-related vulnerability, can result specifically in a striking pattern of parasagittal stripes; and (3) the presence or absence of striped Purkinje cell loss in aged mice is not driven solely by sex or relative age.

## Age-related Purkinje cell loss is due to neurodegeneration and the loss of cells

In mouse models with neurodegenerative ataxia, calbindin and other Purkinje-cell-specific genes and proteins are downregulated prior to the onset of motor dysfunction and Purkinje cell loss (*Vig et al., 1998*; *Hansen et al., 2013*). This molecular signature raises the possibility that the indication of Purkinje cell loss revealed by antibody staining may in fact be the result of reduced Purkinje cell marker expression rather than neurodegeneration. To distinguish between these possibilities, we tested whether Purkinje cells in aged mice were degenerating. First, we observed common hallmarks of Purkinje cell neurodegeneration, such as thickened axons, axonal torpedoes, and shrunken dendritic arbors, in aged mice with striped Purkinje cell loss (n=8) but not in young mice (n=5; *Figure 2A and B*). Quantification of molecular layer thickness in lobule VIII revealed that aged mice with striped Purkinje cell loss have significantly thinner molecular layers compared to young mice (*Figure 2C*), likely due to the regressed dendrites of degenerating Purkinje cells. Aged mice without Purkinje cell loss have minimal axonal pathology (*Figure 2A*), and the thickness of the molecular layer is comparable to that of young mice (*Figure 2B and C*), suggesting that in aged mice without Purkinje cell loss, Purkinje cells are not undergoing major neurodegeneration. We also confirmed whether the pattern of Purkinje cell loss was the same using two different calbindin antibodies (young n=4, aged n=6; *Figure 2—figure supplement 1*).

Second, to verify that the loss of Purkinje cells was not due to downregulation of calbindin specifically, we used Neutral Red, a dye that stains lysosomes, to label surviving cells in cerebellar tissue (*Brandenburg and Blatt, 2022*). Adjacent tissue sections were immunostained for calbindin and/or stained with Neutral Red to locate regions with striped Purkinje cell loss. In the aged mice (n=6), regions of the Purkinje cell and molecular layers without calbindin staining, which indicated missing Purkinje cells, also lacked Neutral Red-positive cell bodies, which based on their morphology were easily distinguished from other cell types by their size and position in young mice (n=6; *Figure 2D*).

Third, to verify the presence of striped Purkinje cell degeneration during normal aging, we used adult Purkinje-cell-specific fluorescent reporter mice, which express a fluorescent reporter specifically in all Purkinje cells because the reporter is driven by *Pcp2^Cre^*. These mice present two advantages for visualizing age-related Purkinje cell loss: (1) Cre expression begins on E17 and continues until all Purkinje cells express Cre in adulthood (*Lewis et al., 2004*), meaning that even if *Pcp2* gene and protein expression are downregulated in advanced age (*Cooper et al., 2024*) or prior to neurodegeneration (*Hansen et al., 2013*), reporter expression will remain constant, as it would have already been activated in all Purkinje cells, and its perdurance would allow continued marking of the recombined cells; and (2) Purkinje-cell-specific reporter expression allows for the visualization of Purkinje

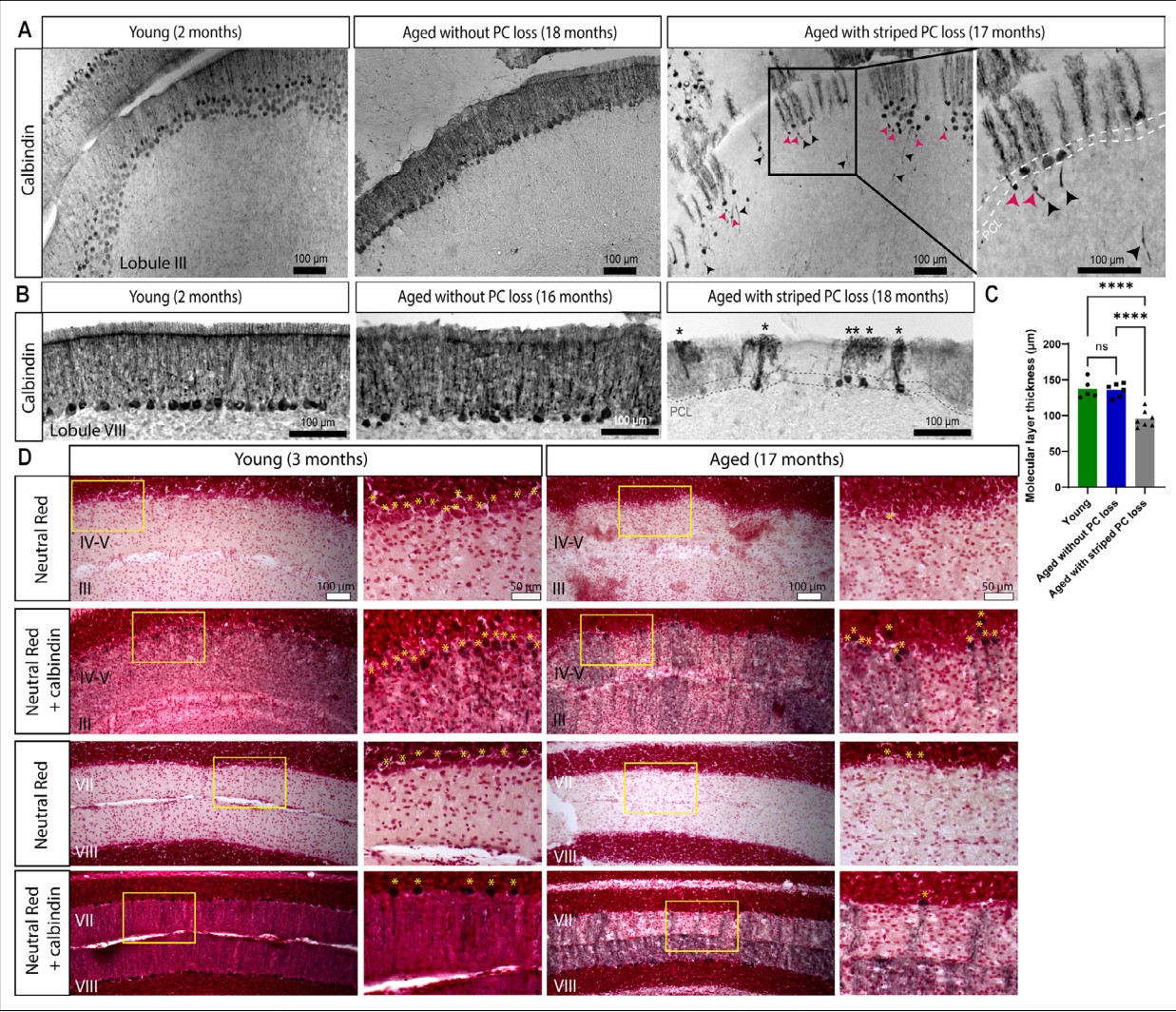

**Figure 2.** Aged mice with striped Purkinje cell loss show cellular level anatomical hallmarks of degeneration. (**A**) Coronal cut cerebellar tissue sections of lobule III immunostained for calbindin. Black arrowheads indicate thickened axons, and pink arrowheads indicate axonal torpedoes. Dashed lines indicate the Purkinje cell layer (PCL). Scale bar = 100 µm. (**B**) Coronal cut cerebellar tissue sections of lobule VIII immunostained for calbindin. Asterisks indicate shrunken dendritic arbors. Scale bar = 100 µm. (**C**) Quantification of molecular layer thickness in lobule VIII; **** indicates p≤0.0001. (**D**) Coronal cut cerebellar tissue sections either stained with Neutral Red or immunostained for calbindin and stained with Neutral Red. Asterisks indicate Purkinje cell bodies. Scale bar = 100 µm; inset scale bar = 50 µm.

The online version of this article includes the following source data and figure supplement(s) for figure 2:

**Source data 1.** Source data for *Figure 2*.

**Figure supplement 1.** Different calbindin antibodies reveal matching expression patterns, including age-related striped Purkinje cell loss.

**Figure supplement 2.** Patterned calbindin expression and staining artifacts can obscure Purkinje cell loss due to neurodegeneration.

cells without relying on calbindin, which can be downregulated with advanced age (*Cooper et al., 2024*; *Iacopino and Christakos, 1990*; *Iacopino et al., 1990*) or prior to neurodegeneration (*Vig et al., 1998*; *Hansen et al., 2013*). We found that the cerebella of young Purkinje-cell-specific fluorescent reporter mice (n=4) displayed uniform reporter expression across the surface, whereas the cerebella of aged Purkinje-cell-specific fluorescent reporter mice (n=17) displayed reporter expression in alternating parasagittal stripes of greater and lesser intensity (*Figure 3A and B*). Furthermore, middle-aged mice (11–15 months old; n=4) tended to have less pronounced bands compared to mice 16 months and older, which had clearer, more widespread bands (*Figure 3A and B*), suggesting a progression of Purkinje cell loss with advanced age. The cerebella of middle-aged mice have striped reporter expression in the anterior zone, large regions with reduced reporter expression in the medial

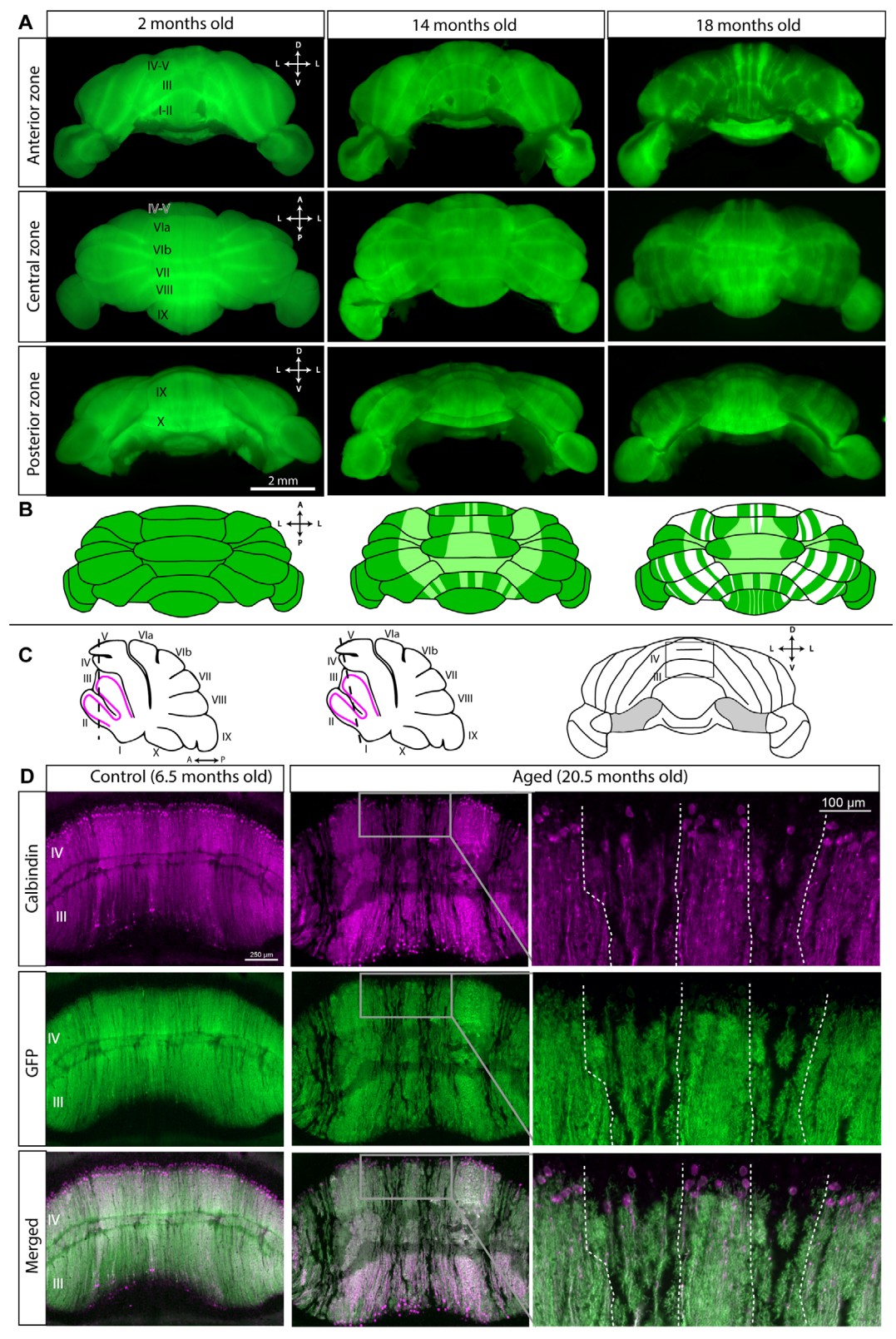

**Figure 3.** Cerebella of aged Purkinje-cell-specific fluorescent reporter mice display the same pattern of Purkinje cell loss as revealed by whole-mount calbindin immunohistochemistry. (**A**) Whole-mount cerebella of Purkinje-cell-specific fluorescent reporter mice visualized with blue light and viewed from different angles. Cerebellar lobules are labeled with Roman numerals. D=dorsal; L=lateral; V=ventral; A=anterior; P=posterior. Scale bar = 2 mm. (**B**) Schematics of the dorsal view of cerebella from young, middle-aged, and older Purkinje-cell-specific fluorescent reporter mice. Lighter colors

*Figure 3 continued on next page*

*Figure 3 continued*

indicate less intense reporter expression. (**C**) Schematics of sagittal sections of the cerebellum and a whole-mount cerebellum indicating the location of tissue sections. Dashed lines indicate the position and angle of tissue sections. (**D**) Coronal cut cerebellar tissue sections of Purkinje-cell-specific fluorescent reporter mice immunostained for calbindin and GFP. Dashed lines indicate boundaries between surviving Purkinje cells and degenerating Purkinje cells. Scale bar = 250 μm; inset scale bar = 100 μm.

The online version of this article includes the following source data and figure supplement(s) for figure 3:

**Figure supplement 1.** Calbindin and GFP reveal overlapping patterns of expression in aged mice with striped Purkinje cell loss.

**Figure supplement 1—source data 1.** Source data for *Figure 3—figure supplement 1*.

vermis and paravermis, and alternating bands in lobule VIII (*Figure 3A and B*), suggesting that age-related Purkinje cell loss may begin in these regions before spreading with advancing age. To confirm that the Purkinje-cell-specific reporter expression and the calbindin whole-mount staining reflected the same pattern, we co-stained coronal cerebellar tissue sections of Purkinje-cell-specific fluorescent reporter mice for calbindin and GFP. In young mice, both calbindin and GFP were expressed in all Purkinje cells (n=3), and in aged mice, calbindin and GFP were expressed in identical, overlapping stripes (n=5; *Figure 3D*, *Figure 3—figure supplement 1*) that reflected the pattern of age-related Purkinje cell loss. These data indicate that the pattern of age-related Purkinje cell loss we revealed with calbindin whole-mount staining matches the pattern that we next observed with reporter expression in Purkinje-cell-specific fluorescent reporter mice.

A combination of dynamic calbindin expression and staining artifacts can affect the visualization of Purkinje cells even in the absence of Purkinje cell loss. We observed striped calbindin expression in both young and aged C57Bl/J6 control mice (*Figure 2—figure supplement 2A*), possibly due to calcium dynamics. We confirmed that these stripes were not due to widespread Purkinje cell loss by staining adjacent tissue sections with Neutral Red, which revealed that the Purkinje cell bodies were still present (*Figure 2—figure supplement 2A*). In addition, in young and aged cerebellar tissue from C57Bl/J6 mice, the Purkinje cell dendrites span the molecular layer (*Figure 2—figure supplement 2A*), whereas in degenerating tissue, shrunken dendrites, thickened axons, and torpedoes can be observed (*Figure 2A and B*). Previous studies have shown that calbindin mRNA and protein are reduced in the cerebella of aged mice, rats, and humans (*Iacopino and Christakos, 1990*; *Iacopino et al., 1990*), and reduced calbindin immunoreactivity has been observed in surviving Purkinje cells in aged rats and in patients with spinocerebellar degeneration (*Amenta et al., 1994*; *Ishikawa et al., 1995*). Therefore, we argue that calbindin expression alone is not a reliable, sufficient indicator of Purkinje cell loss and should be supplemented with other histological and labeling techniques. Staining artifacts can also give the false appearance of Purkinje cell absence, but background staining or Neutral Red can reveal the Purkinje cells (*Figure 2—figure supplement 2B*). For this reason, we used multiple methods to visualize Purkinje cell degeneration and loss. Degenerative Purkinje cell pathology, multiple antibodies, Neutral Red staining, and a Purkinje-cell-specific genetically driven fluorescent reporter confirmed that the striped cellular pattern we observed indicates robust regional Purkinje cell loss in aged mice and that it arises due to neurodegeneration in a subpopulation of Purkinje cells.

## The pattern of age-related Purkinje cell loss overlaps with but is distinct from the overall pattern of zebrin II expression

In mutant mice with Purkinje cell loss (*leaner Fletcher et al., 1996*), *nervous* (*Edwards et al., 1994*), *BALB/c npc^{nih}* (*Sarna et al., 2003*), *C57BLKS/J spm* (*Sarna et al., 2003*), *Cacna1a* null (*Fletcher et al., 2001*), and acid sphingomyelinase knockout (ASMKO) mice (*Sarna et al., 2001*), as well as mice with global brain ischemia (*Welsh et al., 2002*), neurodegeneration occurs according to the expression of zebrin II (*Sarna and Hawkes, 2003*). Zebrin II (an antigen on the aldolase C protein *Ahn et al., 1994*) is the most well-studied cerebellar patterning marker. In vertebrates, zebrin II expression reveals a striking pattern of parasagittal stripes across the cerebellar cortex (*Sillitoe et al., 2005*). Degeneration respects two main populations in the map as revealed by zebrin II expression; for example, in the *nervous* mutant mouse, Purkinje cell loss occurs selectively in zebrin II-positive Purkinje cells (*Edwards et al., 1994*), whereas in models of Niemann-Pick type C disease, zebrin II-negative Purkinje cells die first (*Sarna et al., 2003*). Aged mice displayed three distinct stripes of surviving Purkinje cells in the

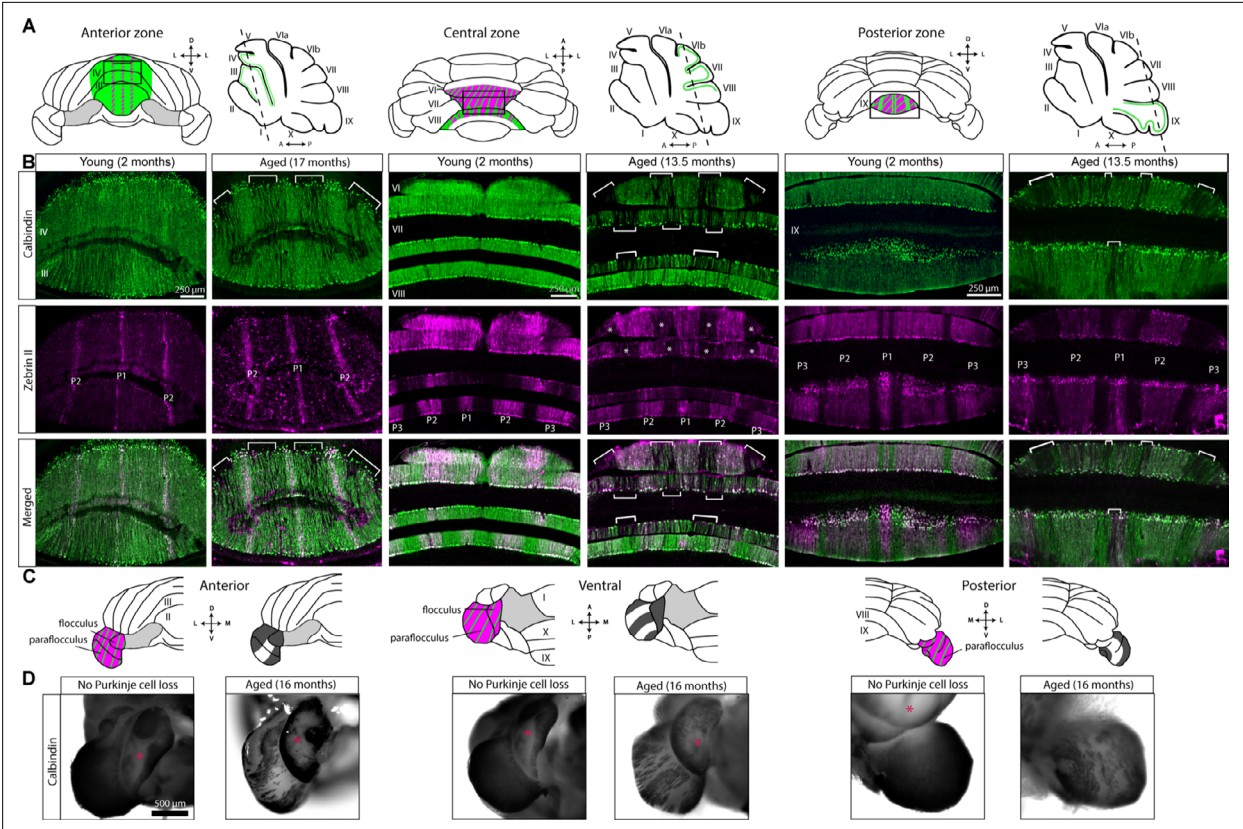

**Figure 4.** The pattern of age-related Purkinje cell loss has some similarities to zebrin II expression, but the unique overall map represents a greater cerebellar complexity. (**A**) Schematics of whole-mount cerebella and sagittal sections of the cerebellum indicating the location of tissue sections. Green indicates calbindin expression, and alternating green and magenta indicates where calbindin and zebrin II are co-expressed. Dashed lines indicate the position and angle of tissue sections. Cerebellar lobules are labeled with Roman numerals. (**B**) Coronal cut tissue sections co-stained for calbindin (green) and zebrin II (magenta). Zebrin II-positive stripes are indicated by P1, P2, and P3. Brackets indicate bands of degenerating Purkinje cells, and asterisks indicate bands of degenerating Purkinje cells in uniformly zebrin II-positive regions. Scale bar = 250 μm. (**C**) Schematics of half of a whole-mount cerebellum indicating calbindin and zebrin II expression (alternating green and magenta) and bands of surviving Purkinje cells as indicated by calbindin expression (dark gray). (**D**) Paraflocculi of whole-mount cerebella immunostained for calbindin and viewed from different angles. Cerebellar lobules are labeled with Roman numerals. D=dorsal; L=lateral; M=medial; V=ventral; A=anterior; *P*=posterior. Scale bar = 500 μm.

The online version of this article includes the following figure supplement(s) for figure 4:

**Figure supplement 1.** Calbindin and Purkinje cell-specific fluorescent reporter expression reflect the same stripes of Purkinje cell loss with respect to zebrin II.

anterior vermis (*Figure 1G–I*), a pathology that resembles the pattern of zebrin II expression in young mice. Therefore, we asked whether the pattern of age-related Purkinje cell loss and survival indeed reflects the pattern of zebrin II expression. To test this, we cut coronal cerebellar tissue sections from aged mice (n=5) and immunostained them for calbindin (to label surviving Purkinje cells) and zebrin II (to reveal Purkinje cell stripe patterning), using cerebellar tissue from young mice (n=4) as controls.

In the anterior zone, age-related Purkinje cell loss respects the zebrin II boundaries. For example, in lobules III and IV, Purkinje cells began to degenerate selectively in zebrin II-negative stripes P1- and P2-, while the zebrin II-positive stripes remained intact (*Figure 4B*). In this region, calbindin and Purkinje-cell-specific reporter expression both reflect the same pattern of Purkinje cell loss with respect to zebrin II (*Figure 4—figure supplement 1*). In other regions, Purkinje cell loss forms clear stripes despite uniform zebrin II expression; for example, in lobule VI and anterior lobule VII, which are almost entirely zebrin II-positive, Purkinje cells degenerate in stripes (*Figure 4B*). This is in contrast to lobule VIII in the same mouse, which is unaffected by Purkinje cell degeneration despite the striped zebrin II expression in this lobule (*Figure 4B*). Interestingly, we observed differences in the relationship between age-related Purkinje cell loss and zebrin II expression within a single lobule. In dorsal lobule IX, Purkinje cell loss occurs in zebrin II-negative stripes, whereas in ventral lobule IX, Purkinje

cell loss occurs in the medial zebrin II-positive stripe P1+ (*Figure 4B*). The paraflocculi display bands of Purkinje cell loss (*Figure 4D*), but given the heterogeneity of this region, with its developmentally defined Purkinje cell clusters and its stripes of intensely and weakly zebrin II-expressing Purkinje cells, and its complicated morphology (*Fujita et al., 2012*; *Fujita et al., 2014*), more detailed analysis is required to fully understand the relationship between this pattern and zebrin II expression. Taken together, these data show that although the pattern of age-related Purkinje cell loss can correspond with zebrin II expression – for example, in the anterior zone – the underlying pattern that dictates Purkinje cell loss during normal aging is more complicated than a single stripe marker would indicate. Instead, differential vulnerability to age-related neurodegeneration may result from complex interactions between Purkinje cell lineage, gene expression patterns, and specific functional properties. These distinct patterns in aged mice may uncover previously unidentified subsets within uniform zebrin II areas or provide clues into afferent fiber-to-Purkinje cell functional interactions that could influence long-term circuit function and health.

Even during extreme Purkinje cell loss, with few Purkinje cells surviving throughout the cerebellum, a pattern of parasagittal stripes remains visible. This was evident in serial sections taken from a 25-month-old mouse and immunostained for calbindin, revealing the subsets of Purkinje cells that were most resistant to degeneration (*Figure 5*). The tissue sections displayed the same pattern of three parasagittal stripes in vermal lobules II through VI, although the width of the stripes was reduced, sometimes to one or two Purkinje cells per stripe. Most of Crus 1, the flocculi, and the paraflocculi had strong calbindin staining, indicating the presence of surviving Purkinje cells. Ventral lobule IX and lobule X were strikingly well preserved in comparison to the rest of the cerebellum. Even within regions with surviving Purkinje cells, the cells were undergoing degeneration. High magnification images of tissue from the 25-month-old mouse revealed extreme morphological abnormalities in Purkinje cells. Beaded recurrent axon collaterals formed plexuses where Purkinje cell somata likely used to be (*Figure 6A–E*), and Purkinje cell dendrites were thickened and fractured (*Figure 6A and C*). We also observed a putative recurrent axon collateral that extended to the top of the molecular layer in a large gap between surviving Purkinje cells (*Figure 6E*). These morphological abnormalities may represent the last efforts of surviving Purkinje cells to reside in an aged cerebellum where the majority of Purkinje cells have degenerated.

## Light sheet imaging reveals a pattern of age-related Purkinje cell loss in the cerebellum

Light sheet imaging, when combined with tissue clearing, allows the visualization of labeled cells in multiple dimensions throughout a brain structure. Given the precise regional specificity of age-related Purkinje cell loss, we performed light sheet imaging of the cerebellum of an aged Purkinje-cell-specific fluorescent reporter mouse (*Video 1*) to fully visualize the pattern. Light sheet imaging revealed that lobule X is resistant to Purkinje cell loss during normal aging, similar to our observations of whole-mount cerebella immunostained for calbindin. Lobule X, the flocculi, and the paraflocculi comprise the nodular zone, a largely zebrin II-positive region (*Figure 4C*). The paraflocculi showed stripes of Purkinje cell loss during aging, as seen on calbindin-stained whole-mounts (*Figure 4D*). By combining the advantages of a Purkinje-cell-specific fluorescent reporter with the ability to reveal patterns within the core of the cerebellum (which are typically hard to appreciate due to the folding of the cortex), where antibodies do not always penetrate, light sheet imaging provides a complete picture of the pattern of age-related Purkinje cell loss.

Taken together, our results from the whole-mount cerebella, coronal tissue sections, and light sheet imaging of Purkinje-cell-specific reporter expression indicate that despite some similarities, the pattern of age-related Purkinje cell loss is similar but not identical to the expression pattern of zebrin II, a reliable marker that defines the endogenous map of Purkinje cell stripes and zones.

## Motor function is impaired in aged mice compared to young mice

Our results show that Purkinje cell loss during normal aging is extensive and occurs throughout the cerebellum. Despite this, the general motor behavior of our cohort of aged mice was ostensibly normal when the mice were observed in their home cages. To investigate the effect of age-related Purkinje cell loss on motor behavior more closely, we performed a series of behavioral tests, including the accelerating rotarod, the horizontal ladder, and analysis in a tremor monitor, on young and aged

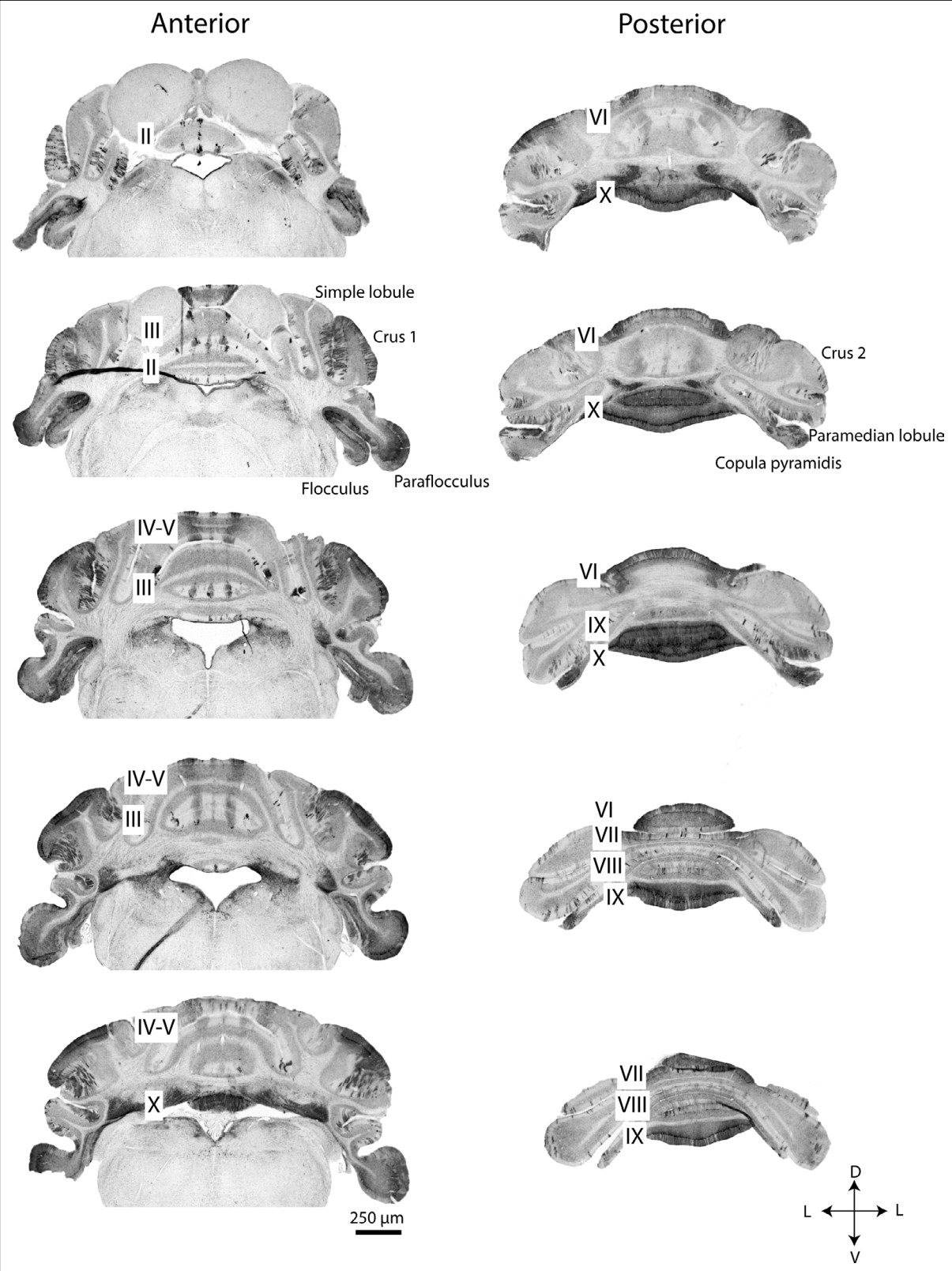

**Figure 5.** Regions with lasting resistance to Purkinje cell loss during normal aging are revealed in serial sections from 25-month-old mice. Coronal cut cerebellar tissue sections immunostained for calbindin and arranged in order from anterior to posterior. Cerebellar lobules are labeled with Roman numerals. D=dorsal; L=lateral; V=ventral; A=anterior; *P*=posterior. Scale bar = 250 μm.

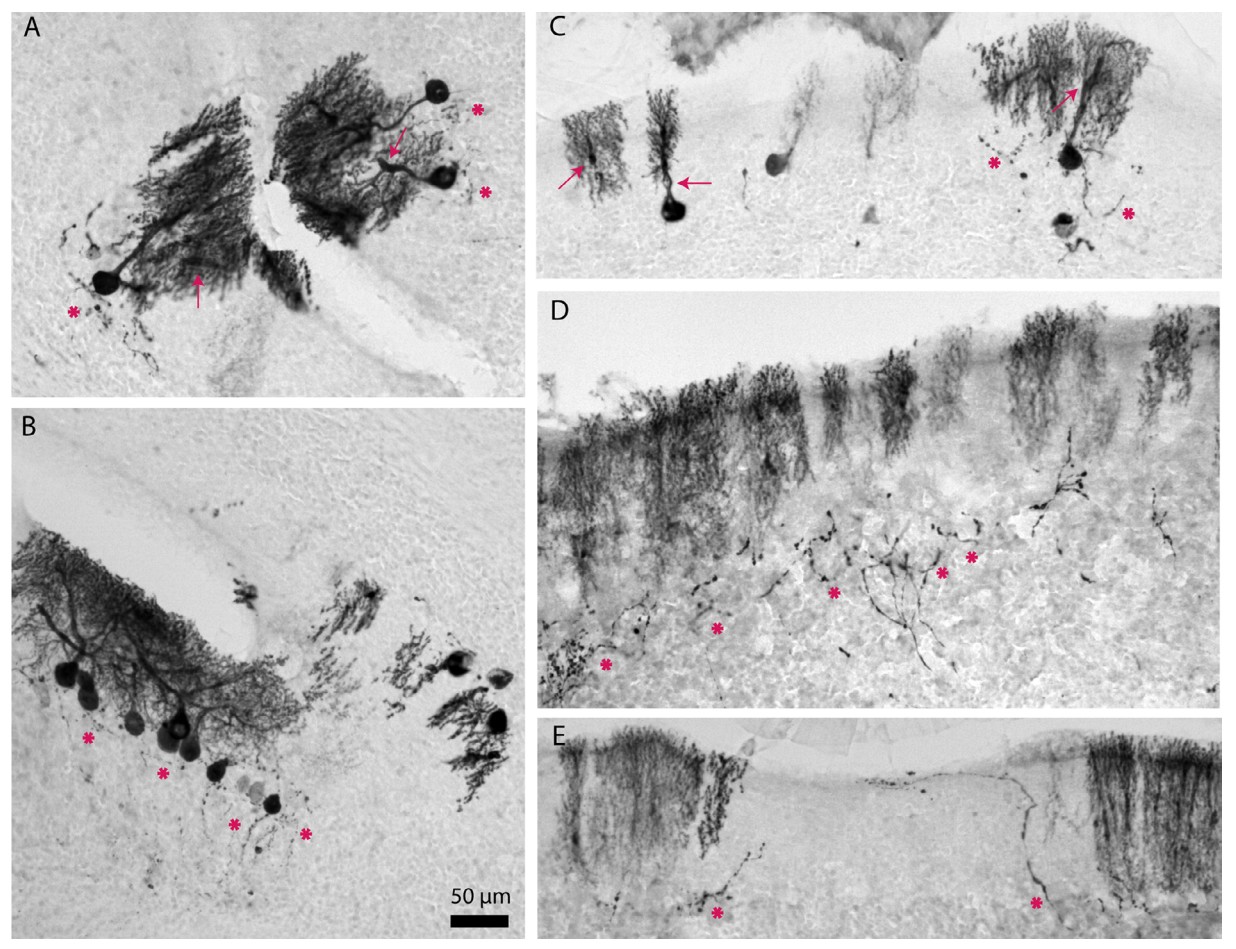

**Figure 6.** Regions with apparent resistance to cell loss in 25-month-old mouse have Purkinje cells with extreme morphological abnormalities. High-magnification images of cerebellar tissue sections immunostained for calbindin. Asterisks indicate recurrent axon collaterals, and arrows indicate thickened dendrites. Scale bar = 50 μm.

mice before sacrificing them and histologically examining the cerebella for Purkinje cell loss. We found that the aged mice (n=12) had a shorter latency to fall from the accelerating rotarod compared to the young mice (n=8; *Figure 7B*). Although the latency to fall was always lower in the aged mice compared to the young mice, the aged mice improved from day to day (*Figure 7B*), demonstrating their continued ability to learn new motor skills.

To test skilled, voluntary movement and limb control, we used a horizontal ladder task. Mice were subjected to an 'easy' trial, where every ladder rung was in place, and a 'difficult' trial, where every other ladder rung was removed. The number of footslips was recorded per trial. As expected, both the young (n=8) and aged (n=12) groups had more footslips during the difficult trial compared to during the easy trial (*Figure 7C*). However, we did not detect a significant difference in the number of footslips between the young and aged groups during either trial (*Figure 7C*).

Because humans and mice have increased tremor with age (*Louis, 2019*; *White et al., 2016*) and the cerebellum is implicated in tremor pathophysiology (*Louis, 2016*; *Handforth,*

**Video 1.** Light sheet imaging reveals the pattern of age-related Purkinje cell loss throughout the cerebellum. Light sheet imaging through the fluorescent reporter cerebellum reveals the pattern of cell loss with excellent anatomical continuity from its surface features to its internal architecture.
https://elifesciences.org/articles/106273/figures#video1

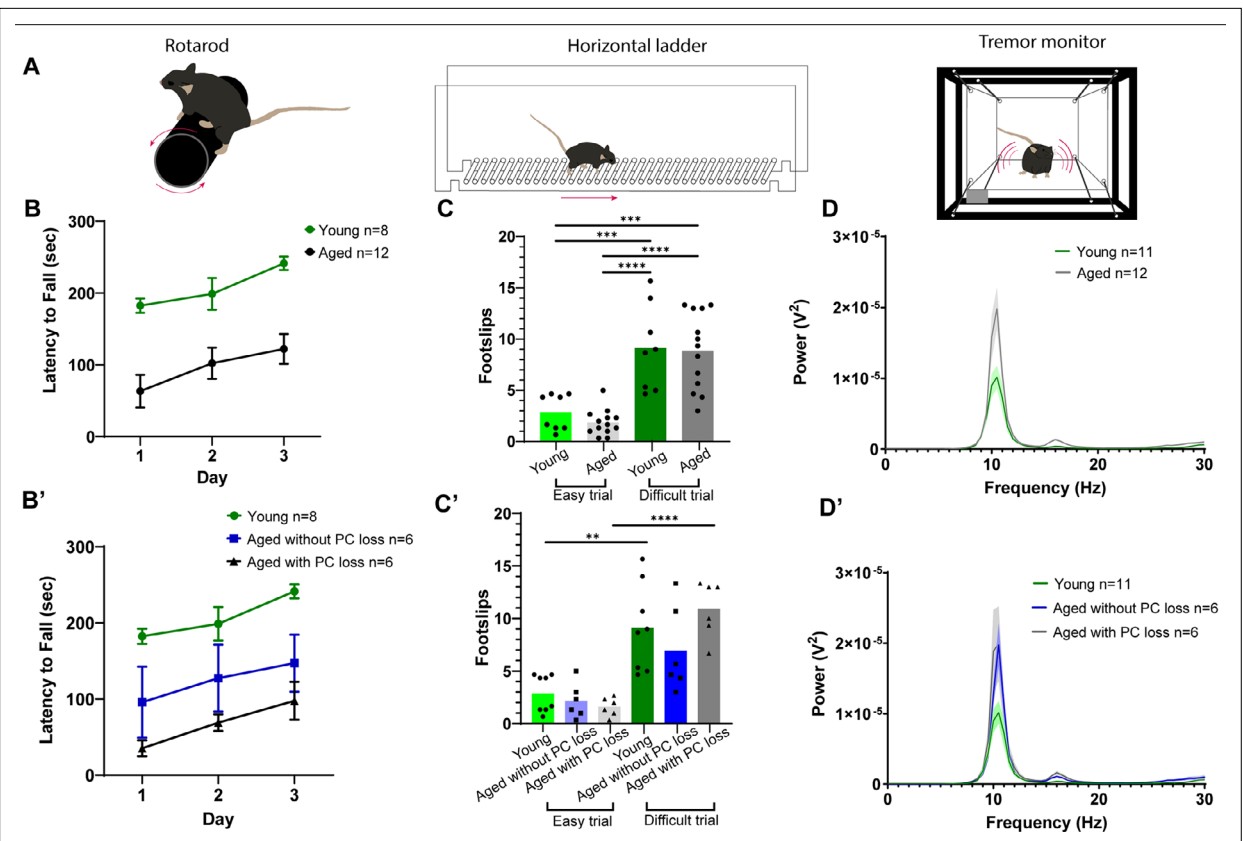

**Figure 7.** Aged mice display impaired performance on the accelerating rotarod and increased tremor but no deficits on the horizontal ladder. (**A**) Schematics of motor function tests. (**B**) Latency to fall from accelerating rotarod. Error bars indicate standard error of the mean. (**B'**) Latency to fall with aged mice sorted based on the presence or absence of Purkinje cell loss. (**C**) Number of footslips when crossing the horizontal ladder. ** indicates p≤0.01, *** indicates p≤0.001, and **** indicates p≤0.0001. (**C'**) Number of footslips with aged mice sorted based on the presence or absence of Purkinje cell loss. (**D**) Power spectrum of tremor detected by a tremor monitor. Error bars indicate standard error of the mean. (**D'**) Power spectrum of tremor with aged mice sorted based on the presence or absence of Purkinje cell loss.

The online version of this article includes the following source data and figure supplement(s) for figure 7:

**Source data 1.** Source data for *Figure 7*.

**Figure supplement 1.** There is no correlation between weight, peak tremor power, or relative age.

**Figure supplement 1—source data 1.** Source data for *Figure 7—figure supplement 1*.

*2016*; *Cerasa and Quattrone, 2016*), we used a custom-built tremor monitor to quantify tremor power and frequency (*Brown et al., 2020*). We found that aged mice (n=12) had significantly higher tremor power compared to young mice (n=11; *Figure 7D*). Peak power in both cohorts occurred at a frequency of ~10 Hz (*Figure 7D*), consistent with instances of pathological tremor being found within the frequency range of physiological tremor in mice (*White et al., 2016*; *Brown et al., 2020*).

We next wondered what structural and/or functional variables might contribute to the motor deficits observed in aged mice, using peak tremor power as an example. We tested whether peak power in aged mice (n=12) was influenced by body weight, relative age, or sex. Female aged mice tended to weigh less and have lower power tremor but still overlapped with male aged mice in terms of weight and peak tremor power (*Figure 7—figure supplement 1A*). We did not find a statistical correlation between peak tremor power, weight, or relative age (defined as the age of an individual mouse in comparison to other mice in the aged group; *Figure 7—figure supplement 1B*). This suggests that although aged mice have significantly increased tremor power compared to young mice (*Figure 7D*), neither weight nor differences in relative age among aged mice contribute significantly to tremor within the aged group. In other words, tremor does not necessarily worsen with increased age beyond a certain point within a given age range. This may be related to our finding that middle-aged mice can have Purkinje cell loss while some older mice do not (*Table 1*). Thus, aging is not a simple linear

process in which increasing age is always negatively (or gradually) correlated with the loss of specific neural circuit functions and a decline in specific behaviors.

Given that not all aged mice have striped Purkinje cell loss, even within the same litter (*Figure 1D–I*; *Table 1*), we wondered whether aged mice without Purkinje cell loss performed better on motor tasks compared to aged mice with Purkinje cell loss. To address this, after behavioral testing, we collected cerebellar tissue sections from the aged mice and immunostained them with calbindin antibody. We subdivided the behavioral data of the aged mice based on whether they had Purkinje cell loss or not. We found that there was no significant difference in rotarod performance, number of footslips on the horizontal ladder, or tremor between aged mice with Purkinje cell loss and aged mice without Purkinje cell loss, though both aged groups had shorter latency to fall on the rotarod and an increased tremor power compared to young mice (*Figure 7B', C' and D'*). Together, these results suggest that while aged mice exhibit abnormalities in tremor and motor coordination, Purkinje cell loss alone, and specifically mild Purkinje cell loss, may not cause these behavioral impairments. Alternatively, Purkinje cell dysfunction, to varying degrees, may set a platform for the development of tremor and motor incoordination in aging mice, which could then co-initiate different abnormal behaviors with a given amount Purkinje cell loss.

## Postmortem tissue from neurologically normal humans reveals age-related Purkinje cell degeneration with the coexistence of healthy and pathological cells

Reports that cerebellar volume is differentially impacted across lobules in human aging (*Raz et al., 1998*; *Han et al., 2020*; *Bernard and Seidler, 2013*; *Luft et al., 1999*; *Hulst et al., 2015*; *Yu et al., 2017*; *Wang et al., 2024*) prompted us to examine human cerebellar tissue at the cellular level to determine whether Purkinje cells exhibit differential vulnerability and resistance to age-related degeneration. We studied postmortem cerebellar tissue from three neurologically normal patients: a 21-year-old, a 57-year-old, and a 74-year-old. Using the calbindin antibody that effectively and reliably labels Purkinje cells in mice, on sagittal cut tissue sections through the vermis, we observed well-preserved Purkinje cells with expansive dendritic arbors in the tissue from the 21-year-old (*Figure 8*). In contrast, the immunostained tissue from the 74-year-old revealed extensive Purkinje cell loss that was observed as noticeable gaps in the Purkinje cell layer that were devoid of somata. In addition, we observed Purkinje cell dendrites with poor integrity and less span in their typically expansive architecture, a change that is indicative of retraction and degeneration of Purkinje cell dendrites (*Figure 8*), similar to our observations of Purkinje cells in aged mice (*Figure 2A and B*). Interestingly, the tissue from the 57-year-old represented an 'intermediate' stage with many robust healthy Purkinje cells that were flanked by areas with Purkinje cell dendrite deterioration and gaps in the Purkinje cell layer

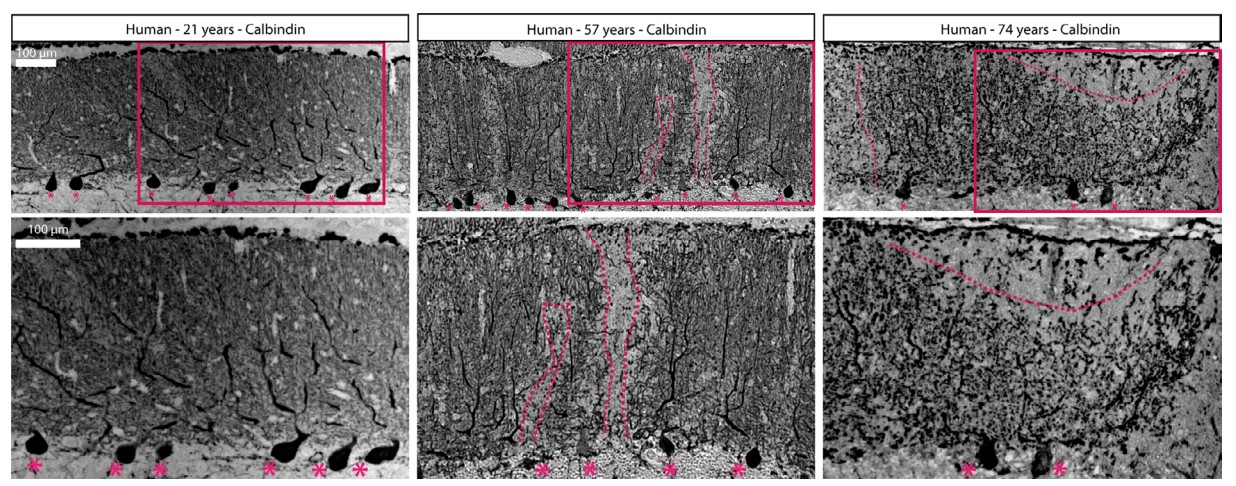

**Figure 8.** Aging humans have Purkinje cell degeneration that can be visualized with calbindin. Asterisks indicate the remaining Purkinje cell bodies. Dashed lines indicate the boundaries of Purkinje cell dendritic arbors. Scale bar = 100 μm.

(*Figure 8*). These data imply the progressive degeneration and loss of Purkinje cells during human aging, as well as differential vulnerability and resistance to such loss among individual cells.

## Discussion

We demonstrate that Purkinje cell loss that occurs during normal aging is not uniform. Instead, Purkinje cell loss occurs in a striking pattern of parasagittal stripes in aged mice. While this striped pattern bears some resemblance to the pattern of zebrin II expression, especially in the anterior zone lobules, the overall pattern is different from zebrin II expression, which typically defines the known stripe patterns of Purkinje cell loss in disease models. Furthermore, we show that aged mice have increased tremor power and deficits in rotarod performance compared to young mice but that performance on the horizontal ladder is preserved. Overall, we have found that age-related Purkinje cell loss occurs in a distinct striped pattern that may provide insight into the selective vulnerability and resistance of cells to neurodegeneration in the normal aging cerebellum.

### Striped Purkinje cell loss during normal aging is more complex than zebrin II expression

Although evidence of striped Purkinje cell loss has not been reported in aged mice prior to our study, striped Purkinje cell loss is widely appreciated in mouse models of disease. In disease models, Purkinje cell loss typically occurs preferentially in either zebrin II-positive (e.g. *nervous* mutant mice *Edwards et al., 1994*) or zebrin II-negative (e.g. BALB/c *np^{nih}* and C57BLKS/J *spm* mice *Sarna et al., 2003*; *Sarna et al., 2001*) Purkinje cells (*Sarna and Hawkes, 2003*). Interestingly, we found that although age-related Purkinje cell loss occurred preferentially in zebrin II-negative stripes in the anterior zone, it followed a unique region-to-region pattern in the rest of the cerebellar cortex. Similar findings, in which Purkinje cell loss occurs in zebrin II-negative stripes in the anterior zone but not the posterior zone, have been observed in a mouse model of autosomal-recessive spastic ataxia of Charlevoix-Saguenay (ARSACS; *Toscano Márquez et al., 2021*), yet the pattern of age-related Purkinje cell loss is distinct. This suggests that other factors in addition to zebrin II identity may influence the selective vulnerability to neurodegeneration. In addition, neurodegeneration is a continuous process. The loss of certain Purkinje cells may trigger the degeneration of nearby cells regardless of regional vulnerability, creating a domino effect that may not precisely reflect zonal markers.

Importantly, zebrin II stripes are not the only example of cerebellar molecular patterning, and therefore, interpretation of patterned Purkinje cell loss should not be limited to the pattern of zebrin II expression alone. There are many molecular markers whose expression patterns form stripes in Purkinje cell subpopulations that correspond with zebrin II stripes (e.g. GABA-B receptor [*Chung et al., 2008*] and PLCβ3 [*Sarna et al., 2006*]), are complementary to zebrin II stripes (e.g. PLCβ4 [*Sarna et al., 2006*]), or have a more complex relationship (e.g. HNK1 [*Eisenman and Hawkes, 1993*], HSP25 [*Duffin et al., 2010*], and NFH [*Demilly et al., 2011*]). In multiple rodent models with Purkinje cell loss, Purkinje cells in the nodular zone that express HSP25 are more resistant to degeneration than HSP25-negative Purkinje cells (*Sarna et al., 2003*; *Sarna et al., 2001*; *Duffin et al., 2010*). In addition to molecular markers, cerebellar zones are also defined by their developmental lineage, climbing fiber and mossy fiber inputs, interneuron organization, and Purkinje cell outputs to the cerebellar nuclei (*Apps and Hawkes, 2009*), any of which might contribute to differential Purkinje cell vulnerability. Therefore, the pattern of Purkinje cell loss during normal aging may reflect a previously unidentified zone marker or a combination of pattern modalities.

Furthermore, we report differences in the susceptibility to Purkinje cell loss not only between Purkinje cell subpopulations but between individual mice. This is evident given that some mice, even mice as old as 18–24 months, do not display Purkinje cell loss (*Table 1*). These aged mice without Purkinje cell loss do not exhibit Purkinje cell pathology indicative of neurodegeneration, although more subtle morphological changes may occur that escape our detection. Especially intriguing is our finding that there are littermate pairs in which one littermate has striped Purkinje cell loss and the other littermate does not. This effect may not be dictated solely by genotype or sex (*Table 1*). Variable phenotypes among littermates are not unusual – for example, graying hair, hair loss, differences in energy levels, posture, metabolism, weight, and body fat content – and this variability can increase with age (*JAX Mice, Clinical & Research Services, 2023*). It is possible that the presence

or absence of age-related striped Purkinje cell loss is another such variable phenotype. This would present an interesting scenario given the striking difference between a cerebellum with no apparent Purkinje cell loss and a cerebellum with broad stripes of degenerated Purkinje cells. Because the pattern adheres to a strict organizational code, this may result in a more binary readout than other metrics that vary with age. One factor that could contribute to this variability is animal housing density. Group-housed mice are more variable in their body compositions compared to singly housed mice (*Nagy et al., 2002*), and group and single housing can each result in different behavioral stressors (aggression among males and social isolation, respectively; *Kappel et al., 2017*; *Gozalo and Elkins, 2023*). These stressors can have behavioral and physiological effects, including neural and endocrine changes (*Kappel et al., 2017*; *Gozalo and Elkins, 2023*), and may indirectly contribute to differential Purkinje cell vulnerability and resilience between individuals during aging.

## A combination of methods confirms patterned Purkinje cell degeneration and loss

Here, we report the presence of reproducible, striped Purkinje cell loss in aged mice, whereas previous studies of aged rodents have concluded that Purkinje cell loss is largely uniform throughout the cerebellum (*Bakalian et al., 1991*; *Hadj-Sahraoui et al., 1996*; *Hadj-Sahraoui et al., 1997*; *Doulazmi et al., 1999*; *Doulazmi et al., 2006*). This discrepancy may be due to the limitations of using tissue sections alone to study patterned neurodegeneration. In previous studies, quantification of age-related Purkinje cell loss has been performed by counting cells in sagittal sections of the cerebellum taken every 320 µm (*Hadj-Sahraoui et al., 1996Hadj-Sahraoui et al., 1997*; *Doulazmi et al., 1999*; *Doulazmi et al., 2006*). To use the pattern of Purkinje cell loss observed in our study as an example, the observations would vary greatly depending on where a sagittal section was taken. For example, a sagittal section taken at the midline could reveal little or no Purkinje cell loss, whereas a section taken more laterally could reveal large swathes of missing Purkinje cells. The full stripes could only be properly and fully appreciated with coronal sections, which risks leaving out information about anterior-posterior dendrite defects that can be better appreciated with sagittal sections. Therefore, relying on only coronal or sagittal sections is likely insufficient to appreciate complex regional differences in Purkinje cell vulnerability during aging. Our study has the advantage of using whole-mount visualization and light sheet imaging in addition to coronal and sagittal tissue sections, enabling the multi-dimensional visualization of cerebellar patterning.

Methods-based discrepancies in regional cerebellar degeneration have also been observed in elderly patients. One imaging study found reduced volume in the hemispheres and vermal lobules VI-X while the anterior vermis was unaffected (*Raz et al., 1998*), but a later study by the same authors found uniform vulnerability across the vermis and attributed the differences to methodological differences (*Raz et al., 2001*), underscoring the importance of factoring technique into results about regional degeneration in the highly compartmentalized cerebellum. However, the majority of studies in elderly patients have found that when cerebellar atrophy is observed, atrophy is most severe in the anterior cerebellum (*Han et al., 2020*; *Andersen et al., 2003*; *Torvik et al., 1986*; *Bernard and Seidler, 2013*; *Hulst et al., 2015*; *Wang et al., 2024*), with significant volume reduction or Purkinje cell loss often observed in the vermis (*Han et al., 2020*; *Sjobeck et al., 1999*; *Luft et al., 1999*; *Yu et al., 2017*). Similarly, we found that aged mouse cerebella with striped Purkinje cell loss had the most profound loss in the anterior cerebellum, including the vermis (*Figure 1A*). This similarity in morphological phenotype, combined with our finding of Purkinje cell loss in cerebellar tissue from middle-aged and older, neurologically normal human patients (*Figure 8*), suggests translatability between our findings of Purkinje cell loss in aged mice and humans. However, a much larger study must be undertaken with a greater number of human tissue specimens, accounting for a wider span of ages, possible gender differences, race, and other person-to-person variabilities such as socioeconomic status, diet, exercise, and overall health. To determine the extent to which our findings in mice relate to human aging, it will also be important to study regional Purkinje cell loss at the cellular level in elderly humans. As the field currently stands, studies of cerebellar atrophy in humans lack the resolution to examine Purkinje cell degeneration at the cellular level, and detailed analysis of human Purkinje cell subpopulations is confounded by the size and complexity of the cerebellum. Studies of human postmortem cerebellum tissue have long neglected the complex organization of the cerebellum, but a recent study used a marker of Purkinje cell stripes in mice to differentiate Purkinje cell subtypes in human tissue,

thereby identifying differential axonal pathology in essential tremor (*Widner et al., 2025*). A similar method could be useful for studying the link between patterned Purkinje cell loss in mice and humans.

## Regional vulnerability creates distinct patterns of cerebellar degeneration in diseases versus during normal aging

In humans, cerebellar degeneration occurs in distinct patterns reflected by the affected lobules in a given subtype of neurodegenerative disease (*Hernandez-Castillo et al., 2018*) and whether the patient is affected by a neurodegenerative disease or normal aging (*Hulst et al., 2015*; *Gellersen et al., 2021*). The pattern of cerebellar degeneration may impact the manifestation of symptoms because of the functional compartmentalization of the cerebellum at the level of lobules and the stripes within these lobules. Purkinje cell synaptic plasticity and firing rates differ depending on the stripes they inhabit (*Cerminara and Apps, 2011*), and modules differ in terms of behavioral function, although these functions likely overlap across modules to some degree (*Apps et al., 2018*). This suggests that cerebellar compartments and Purkinje cell subpopulations may represent a key for unlocking the neural correlates of cerebellar dysfunction in disease and aging.

The mechanisms underlying patterned cerebellar degeneration remain unknown, although evidence points to the differential vulnerability of Purkinje cell subpopulations (*Welsh et al., 2002*; *Duffin et al., 2010*; *Wassef et al., 1987*; *Chung et al., 2016*; *Martin et al., 2019*). One potential mechanism for differential Purkinje cell vulnerability is stripe-specific excitotoxicity. Our study found that in aged mice, where Purkinje cell loss corresponds with zebrin II expression, Purkinje cell loss occurs preferentially in zebrin II-negative stripes. Zebrin II-negative Purkinje cells have a higher average firing frequency than zebrin II-positive Purkinje cells (*Zhou et al., 2014*, *Xiao et al., 2014*), which may make zebrin II-negative Purkinje cells more susceptible to excitotoxicity. Accordingly, excitatory amino acid transporter 4 (EAAT4) expression is restricted to zebrin II-positive Purkinje cells, and deafferentation of Purkinje cells prevents ischemia-induced Purkinje cell loss in zebrin II-negative stripes (*Welsh et al., 2002*). This selective loss of Purkinje cell subpopulations during aging has functional implications. Recent work has shown that Purkinje cells in aged mice have a reduced firing rate compared to Purkinje cells in young mice (*Fields, 2024*). Furthermore, the distribution of Purkinje cell firing frequencies is similar in young and aged mice, but higher-firing Purkinje cells are absent in aged mice (*Fields, 2024*), suggesting that higher-firing Purkinje cells die while lower-firing cells survive. This corresponds with the results of our study, in which higher-firing zebrin II-negative Purkinje cells tend to degenerate before lower-firing zebrin II-positive Purkinje cells during aging. Stripe-specific excitotoxicity, in combination with potentially neuroprotective proteins such as HSP25 (*Armstrong et al., 2011*), may influence differential Purkinje cell vulnerability in aged mice. Understanding the mechanistic origins of differential vulnerability and resistance in distinct cell populations could shed light on therapeutic methods to block or even prevent neurodegeneration in disease and aging.

## Mild age-related Purkinje cell loss alone may not cause overt motor impairments

The aged mice in our study displayed overtly normal motor behavior during observation, while behavioral tests revealed that aged mice had higher power tremor than young mice and impaired performance on the rotarod. These findings are in accordance with previous behavioral studies of normal aged mice (*Shoji and Miyakawa, 2019*; *White et al., 2016*). Interestingly, aged mice performed as well as young mice on the horizontal ladder test, even when the difficulty was increased by removing every other ladder rung (*Figure 7C*). This may be because aged mice can adapt abnormal kinematics for voluntary movements to compensate for gradual functional deficits during the aging process, whereas involuntary, whole-body tasks such as the accelerating rotarod prove more difficult to overcome.

Upon examination of the cerebella, the aged mice used in these behavioral tests had either no Purkinje cell loss or mild Purkinje cell loss with no evident stripes (severe cases would typically be accompanied by a clear striped pattern). To determine the influence of Purkinje cell loss on age-related motor deficits, we separated the behavioral data based on the presence or absence of Purkinje cell loss. We did not find a significant difference in motor behavior between the two aged subgroups, which suggests that mild Purkinje cell loss alone is not sufficient to cause the observed motor deficits. The aging process involves the dysfunction and degeneration of different classes of neurons within different brain regions (with both overlapping and distinct temporal onsets), which

likely all co-contribute to the decline of motor behavior over time. Additionally, it remains to be determined whether the Purkinje cells in aging mice are functionally normal before their degeneration. It is possible that deterioration begins earlier and that they are contributing to network dysfunction well before their eventual degeneration and that all aged mice are vulnerable to motor deficits when Purkinje cells display abnormal physiological properties. In-depth, precise behavioral phenotyping over time will be necessary to investigate a potential link between progressive changes in Purkinje cell morphology and motor function during aging.

We focused this study on mice of the C57BL/6J background. A next step would be to investigate age-related Purkinje cell loss in other inbred and outbred strains. Different strains, including C57BL/6J, display differences in motor behavior tests (*McFadyen et al., 2003*; *Brooks et al., 2004*), as well as differences in cerebellar morphology (*Inouye and Oda, 1980*). Therefore, whether strain-specific patterns of Purkinje cell loss and motor behavior exist in aged mice is an important question for future research.

## Conclusion

We show that Purkinje cell loss occurs in a striped pattern during normal aging. We revealed this patterned neurodegeneration using a combination of whole-mount immunohistochemistry, tissue staining on sections, and transgenic mice encoded with a Purkinje-cell-specific reporter. Our work establishes a fresh perspective for how patterns of degeneration in models of aging and disease could inform symptomology and regional vulnerability. Despite the apparent chaos of widespread Purkinje cell degeneration, the strict organization of the cerebellum established early in development lends an order to the chaos. Future studies will benefit from identifying the causes of differential vulnerability across Purkinje cell subpopulations and between individual mice, as well as investigating Purkinje cell loss and motor function in aged mice of multiple strains. By understanding the mechanisms of patterned, age-related Purkinje cell loss, we can better appreciate the functional implications of neurodegeneration during normal aging. Eventually, the characteristics that confer resistance to neurodegeneration in specific Purkinje cell subpopulations may prove useful for designing effective treatments that maximize features of healthy brain aging.

## Materials and methods

### Mice

Mouse husbandry and experiments were performed under an approved Institutional Animal Care and Use Committee (IACUC) protocol at Baylor College of Medicine (BCM). All mice used in this study were maintained in our colony at BCM. Mice between 2 and 5 months of age were categorized as young mice, mice between 11 and 15 months were considered middle-aged, and mice older than 16 months were considered old (consistent with information at Jackson Laboratory). Both males and females were used. A mixed population of mice was used, some of which were C57BL/6J mice ordered from Jackson Laboratory (#000664) and some of which were multiple generations descended from the following Jackson Laboratory strains: *Pcp2^{Cre}* (#006207), *ROSA26^{lsl-DTR}* (#007900), *Mash-1^{CreER}* (#012882), *Pdx1^{Cre}* (#014647), *Gdnf^{CreER}* (#024948), *VGAT^{fx/fx}* (#12897), *Tau^{lsl-mGFP-lacZ}* (#21162), *ROSA26^{Ai32}* (#24109), and *ROSA26^{Ai40D}* (#021188). No conditional knockout or knock-in crosses were used except for *Pcp2^{Cre};ROSA26^{Ai32}* and *Pcp2^{Cre};ROSA26^{Ai40D}* mice, which were used to fluorescently tag Purkinje cells. *Pcp2^{Cre};ROSA26^{Ai32}* and *Pcp2^{Cre};ROSA26^{Ai40D}* mice were used interchangeably and will be referred to as Purkinje-cell-specific fluorescent reporter mice throughout the study.

### Perfusion and sectioning

Mice were anesthetized by intraperitoneal injection of Avertin (2, 2, 2-Tribromoethanol, Sigma-Aldrich catalog #T4). Cardiac perfusion was performed with 0.1 M phosphate-buffered saline (PBS; pH 7.4), followed by 4% paraformaldehyde (4% PFA) diluted in PBS. Brains were dissected and post-fixed at 4 °C for at least 24 hr in 4% PFA. Brains were then cryoprotected in sucrose solutions (10% sucrose in PBS, then 20% sucrose, then 30% sucrose) and embedded in Tissue-Tek O.C.T. compound (Sakura Finetek USA; catalog #4583). Tissue sections were cut on a cryostat with a thickness of 40 µm and placed in PBS.

## Free-floating tissue section immunohistochemistry

Immunohistochemistry on free-floating frozen-cut tissue sections was performed as described previously (*Brown et al., 2019*). Rabbit anti-calbindin (1:10,000; Swant) or mouse anti-calbindin (1:2000; Sigma) was used to label all Purkinje cells. Mouse anti-zebrin II (1:250; gift from Dr. Richard Hawkes, University of Calgary, Alberta, Canada) was used to reveal Purkinje cell stripes. Immunoreactive complexes were visualized with either 3,3'-diaminobenzidine tetrahydrochloride (DAB, 0.5 mg/mL; Sigma-Aldrich), nickel-DAB (DAB Substrate Kit; Vector Labs), or anti-mouse or anti-rabbit secondary antibodies conjugated to fluorophores (1:1500; Invitrogen). For DAB reactions, horseradish peroxidase (HRP)-conjugated goat anti-rabbit and goat anti-mouse secondary antibodies (diluted 1:200 in PBS; DAKO) were used to bind the primary antibodies. Antibody binding was revealed by incubating the tissue in the DAB solution, which was made by either dissolving a 100 mg DAB tablet in 40 mL PBS and 10 µL 30% $H_2O_2$ or using the DAB Substrate Kit. The DAB reaction was stopped with PBS when the optimal color intensity was reached.

After staining, tissue sections were placed on electrostatically coated glass slides. Tissue sections were coverslipped using either Cytoseal (for DAB) or FluoroGel with Tris buffer (for immunofluorescence; Electron Microscopy Sciences).

## Neutral Red

After tissue sections were placed on electrostatically coated glass slides, they were left to dry overnight. Then, slides were dipped briefly in distilled water before being immersed in 1% Neutral Red for 30 min. Next, slides were subjected to an ethanol series and xylene before being coverslipped with Cytoseal.

## Whole-mount immunohistochemistry

Whole-mount immunohistochemistry was performed as previously described (*Sillitoe and Hawkes, 2002*; *White et al., 2012*). Cerebella were post-fixed in Dent's fixative for 6 hr at room temperature (RT) and bleached in Dent's bleach overnight at 4 °C. The next day, cerebella were dehydrated in two 30 min rounds of methanol (MeOH) at RT. The cerebella were then subjected to five freeze/thaw cycles before being placed at –80 °C overnight. The next day, the cerebella were rehydrated with washes in 50% MeOH/50% PBS, 15% MeOH/85% PBS, and 100% PBS for 1 hr each at RT. The tissue was enzymatically digested with 10 ug/mL Proteinase K in PBS for 3 min. Then, the cerebella were washed in PBS three times for 10 min each at RT. Tissue was blocked in PBSMT overnight at 4 °C. The next day, the tissue was incubated in primary antibodies diluted in PBSMT with 5% DMSO for 48 hr at 4 °C. After incubation, the cerebella were washed in PBSMT twice for 2 hr each at 4 °C. Then, the tissue was incubated in secondary antibodies diluted in PBSMT with 5% DMSO at 4 °C overnight. The next day, the cerebella were washed in PBSMT twice for 2 hr each at 4 °C, followed by a single wash in PBT for 1 hr. Finally, the cerebella were incubated in DAB solution for 10 min. The DAB reaction was stopped by placing the cerebella in 0.04% sodium azide.

## Microscopy

Images of stained tissue sections were captured with either a Zeiss AxioCam MRm (fluorescence) camera mounted on a Zeiss Axio Imager.M2 microscope or a Leica DMC2900 (brightfield) camera mounted on a Leica DM4000 B LED microscope. Whole-mount cerebella immunostained for calbindin were placed in 1% agar and immersed in PBS, then imaged with a Zeiss AxioCam MRc 5 camera mounted on a Zeiss Axio Zoom.V16 microscope. Brains from Purkinje cell-specific fluorescent reporter mice were imaged with a Zeiss AxioCam MRm camera mounted on a Zeiss Axio Zoom.V16 microscope immediately after perfusion. After imaging, the raw data was imported into Adobe Photoshop, which was used to correct brightness and contrast levels. Schematics were created in Adobe Illustrator.

## Clearing and light sheet imaging

Brains were cleared with the EZ Clear method as described previously (*Hsu et al., 2022*). Whole brain images were acquired with a Zeiss Light Sheet Z1 at a refractive index of 1.52 with a 5 x objective. Image tiles were stitched together with Stitchy and visualized with Arivis.

## Accelerating rotarod

The rotarod (ENV-571M, Med Associates, Inc, Vermont, USA) was set to accelerate from 4 to 40 rpm in 5 min (setting 9) and was stopped at 300 s if the mice successfully stayed on for this duration. Mice

rested for at least 10 min between trials. Rotarod performance was measured in three trials per day for three consecutive days.

## Tremor monitor

Tremor was measured using a custom-built model similar to that described previously (*Brown et al., 2020*). Mice were placed in a translucent plastic box with an open top. The box was held steady in the air by eight elastic cords attached to corners of the box and to a scaffold. An accelerometer at the bottom of the box detected movements of the box. Signals from the accelerometer were recorded and analyzed in Spike2 software. Power spectrums of tremor across naturally occurring tremor frequencies (0–30 Hz) were made using a fast Fourier transform (FFT) with a Hanning window. An offset was applied to center the tremor waveform on 0, and the recordings were downsampled to produce frequency bins aligned to whole numbers. For each mouse, the first 120 s of recording in the tremor monitor was defined as the acclimation period, and the following 180 s of recording was used for analysis.

## Horizontal ladder

The horizontal ladder test was performed on a custom-built ladder consisting of rods placed horizontally between two plexiglass walls. Mice were placed at the entrance of the ladder and allowed to walk across. If a mouse turned around before completing the test, the mouse was placed back at the entrance of the ladder to restart the trial. Video recordings of each trial were analyzed to count the number of footslips per 50 cm section of ladder. A footslip was counted when a foot passed below the level of the rungs. The easy horizontal ladder test involved rods spaced 1 cm apart, and the difficult test involved the removal of every other rod, resulting in rods spaced 2 cm apart. Each mouse completed three trials of the easy test in one day, followed by three trials of the difficult test the next day.

## Human tissue

Use of human postmortem brain tissue was granted exemption by the Baylor College of Medicine Institutional Review Board. All procedures involving a human participant were performed in accordance with the National Research Committee and the 1964 Declaration of Helsinki and its later amendments or comparable ethical standards. The three brains were removed as part of routine hospital autopsy with no significant neurological history or neuropathological findings. Age was extracted from the autopsy report.

## Quantitative analyses and statistics

Fisher's exact test was performed to determine whether the presence of striped Purkinje cell loss in aged mice is sex dependent. Mice were grouped by sex and by whether striped Purkinje cell loss was observed.

Pixel intensity of calbindin and GFP was calculated with the plot profile function in ImageJ (version 1.54 g). Plot profile was performed on identically sized rectangular regions of interest that were placed over lobules II and III. One anterior coronal tissue section immunostained for calbindin or calbindin plus GFP was analyzed per mouse. Because pixel intensity is a measure of brightness, immunofluorescence has a high pixel intensity, whereas the DAB chromogen has a low pixel intensity. For this reason, pixel intensity was used to plot mediolateral differences in immunofluorescence, and inverse pixel intensity was used to plot mediolateral differences on tissue sections where DAB was used to visualize the staining.

Molecular layer thickness was calculated by measuring the molecular layer in dorsal lobule VIII in coronal tissue sections immunostained with calbindin DAB. We chose to measure lobule VIII because of its striking zebrin II pattern and because the cerebellar cortex of this lobule has regions without curvature that are ideal for measuring. One tissue section was used per mouse, and three measurements of the molecular layer were taken per tissue section (midline, left of midline, and right of midline). The three measurements in each tissue section were averaged, and the averages were plotted. The data was analyzed with one-way ANOVA with multiple comparisons.

For the accelerating rotarod and horizontal ladder tests, the three trials per day were averaged, and the averages were plotted. The horizontal ladder data was analyzed with one-way ANOVA with Tukey's multiple comparisons test.

## Acknowledgements

This work was supported by Baylor College of Medicine, Texas Children's Hospital, the Jan and Dan Duncan Neurological Research Institute (Texas Children's Hospital Duncan NRI), the National Institute of Neurological Disorders and Stroke (RVS: R01NS119301 and R01NS127435; SGD: F31NS129279), Eunice Kennedy Shriver National Institute of Child Health and Human Development of the National Institutes of Health under Award Number P50HD103555 for use of the Cell and Tissue Pathogenesis Core (the BCM IDDRC), and the BCM Optical Imaging and Vital Microscopy Core, with the expert assistance of Chih-Wei Hsu. Roy V Sillitoe is supported by the Ting Tsung and Wei Fong Chao Foundation. The content is solely the responsibility of the authors and does not necessarily represent the official views of the National Center for Research Resources or the National Institutes of Health.

## Additional information

### Competing interests

Roy V Sillitoe: Reviewing editor, eLife. The other authors declare that no competing interests exist.

### Funding

| Funder | Grant reference number | Author |
| --- | --- | --- |
| National Institute of Neurological Disorders and Stroke | R01NS119301 | Roy V Sillitoe |
| National Institute of Neurological Disorders and Stroke | R01NS127435 | Roy V Sillitoe |
| National Institute of Neurological Disorders and Stroke | F31NS129279 | Sarah G Donofrio |
| Eunice Kennedy Shriver National Institute of Child Health and Human Development | P50HD103555 | Roy V Sillitoe |

The funders had no role in study design, data collection and interpretation, or the decision to submit the work for publication.

### Author contributions

Sarah G Donofrio, Conceptualization, Data curation, Formal analysis, Funding acquisition, Validation, Investigation, Visualization, Methodology, Writing – original draft, Writing – review and editing; Cheryl Brandenburg, Conceptualization, Formal analysis, Investigation, Visualization, Methodology, Writing – review and editing; Amanda M Brown, Resources, Formal analysis, Methodology, Writing – review and editing; Tao Lin, Investigation, Methodology, Writing – review and editing; Hsiang-Chih Lu, Resources, Methodology, Writing – review and editing; Roy V Sillitoe, Conceptualization, Resources, Formal analysis, Supervision, Funding acquisition, Investigation, Writing – original draft, Project administration, Writing – review and editing

### Author ORCIDs

Sarah G Donofrio ⓘ https://orcid.org/0000-0003-1680-3302
Cheryl Brandenburg ⓘ https://orcid.org/0000-0002-9370-2313
Amanda M Brown ⓘ https://orcid.org/0000-0002-1484-8972
Roy V Sillitoe ⓘ https://orcid.org/0000-0002-6177-6190

### Ethics

Mice were housed in an AAALAS-certified animal facility. All procedures to maintain and use these mice were approved by the Institutional Animal Care and Use Committee for Baylor College of Medicine (animal protocol number AN-5996).

Reviewer #1 (Public review): https://doi.org/10.7554/eLife.106273.3.sa1
Reviewer #2 (Public review): https://doi.org/10.7554/eLife.106273.3.sa2
Reviewer #3 (Public review): https://doi.org/10.7554/eLife.106273.3.sa3
Author response https://doi.org/10.7554/eLife.106273.3.sa4

## Additional files

### Supplementary files
MDAR checklist

### Data availability
All data generated or analyzed during this study are included in the manuscript and supporting files; source data files have been provided.

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
