## [Editor Report · eLife Assessment]

This **important** study presents findings on the patterned loss of Purkinje cells in the cerebellum during aging. The **compelling** data nicely support the conclusions of this study. This work advances understanding of mechanisms underlying neurodegeneration with aging and provides the basis for development of treatments for age-related neurological disorders.

---

## [Referee Report · Reviewer #1 (Public review)]

Summary:

In this study, Donofrio et al. investigated cerebellar Purkinje cell (PC) degeneration during normal aging using both mouse and human samples. They found that PC loss followed a stripe pattern rather than occurring randomly. Although this pattern resembled the pattern of zebrin II expression in the anterior cerebellum, the overall pattern was different from zebrin II expression. Surviving PCs exhibited severe degeneration, including thickened axons, axonal torpedoes and shrunken dendrites. These structural changes were accompanied by functional deficits in motor coordination and tremor. Understanding why certain PC subpopulations are more vulnerable than others may provide insight into regional susceptibility (or resilience) to aging and inform potential therapeutic strategies for age-related neurological disorders. Overall, the findings are novel and significant, supported by compelling evidence from structural and functional analyses. The authors have fully addressed my previous concerns and improved the clarity of their presentation. I believe this work will have a significant impact in the field.

---

## [Referee Report · Reviewer #2 (Public review)]

Summary:

The cerebellum is known to be vulnerable to aging, yet specific cell type vulnerability remains understudied. This important study convincingly demonstrate that the normal aged mouse cerebellum exhibits Purkinje cell loss, and that the vulnerable PCs to age are arranged on the basis of known Zebrin stripe pattern that represents a particular subtype of the PCs. As the authors wrote, future studies should investigate why this PC loss phenotype occurs stochastically across the population, and whether these findings parallel human cerebellar aging.

Strength:

• Banding pattern of PC loss is very clearly demonstrated by combining immunostaining for Zebrin.

• A critical methodological concern that a standard PC marker, Calbindin, could be compromised in aging has been addressed by performing control experiments with appropriate counterstaining and a transgenic mouse.

• Parallels with neurodegenerative phenotype would be helpful to understand the mechanisms of age-related PC loss in future.

Weakness:

• Limited strain diversity: The study exclusively uses C57BL/6J mice despite known genetic and motor differences among even closely related strains like C57BL/6N, weakening the generalizability of the findings. However, on the other hand, the presence of age-related PC loss makes C57BL/6J an interesting mouse model for studying aging of the cerebellum.

• Linkages with normal human aging and cerebellar function is not supported well. It remains unclear whether this PC loss phenomenon is universal or specific to a particular individual, and whether specific to human PC subtype.

---

## [Referee Report · Reviewer #3 (Public review)]

Donofrio et al. report a new observation that in normal aging mice, anti-calbindin whole-mount staining and coronal immunohistochemistry in the cerebellum often show a sagittally patterned loss of Purkinje cells with age. The authors address a central concern that calbindin antibody staining alone is not sufficient to definitively assess Purkinje cell loss, and corroborate their antibody staining data with transgenic Pcp2-CRE x flox-GFP reporter mice and Neutral Red staining. The authors then investigate whether this patterned Purkinje loss correlates with the known parasagittal expression of zebrin-II, finding a strong but imperfect correlation with zebrin-II antibody staining. They next draw a connection between this age-related Purkinje loss to the age-related decline in motor function in mice, with trending but non-significant statistical association between the severity/patterning of Purkinje loss and motor phenotypes within cohorts of aged mice. Finally, the authors look at post-mortem human cerebellar tissues from deceased healthy donors between 21 and 74 years of age, finding a positive correlation between Purkinje degeneration and age, but with unknown spatial patterning.

The conclusions drawn from this study are well supported by the data provided, with image quantification corroborating visual observations. The authors highlight several examples of parasagittal patterning of Purkinje cell degeneration in disease, and they show that proper methodologies must be used to account for these patterns to avoid highly variable data in the sagittal plane. The authors aptly point out that additional work is needed to investigate the spatial patterns of Purkinje cell loss in the human cerebellum.

---

## [Author Response]

The following is the authors’ response to the original reviews.

**Reviewer #1 (Public review):**
While the authors have largely ruled out zebrin II as the key protein underlying PC vulnerability or resistance to age-related loss, the molecular basis of this phenomenon remains unidentified. This reviewer acknowledges the complexity of this investigation and considers it a minor issue, as the manuscript thoughtfully discusses the gap and highlights it as a future direction.

We appreciate the reviewer’s acknowledgement of the complexity of determining the molecular basis of differential Purkinje cell vulnerability. Moreover, we acknowledge that zebrin II expression/identity is not the only factor in determining vulnerability; rather, the compartmentalized map as a whole may dictate these differences. We are eager to shed light on this issue through future study.

In cases where no PC loss is observed in aged mice (Figure 1F), it is unclear whether these PCs undergo morphological degeneration, such as thickened axons and shrunken dendrites. Further characterization of these resilient PCs would help understand why the aged mice without PC loss still exhibit motor deficits (Figure 7).

Thank you for the excellent idea of examining Purkinje cell morphology in aged mice without Purkinje cell loss. Upon looking for hallmarks of neurodegeneration, such as shrunken dendrites and axonal swellings, in aged mice without Purkinje cell loss, we observed minimal axonal pathology and no shrinkage of the molecular layer. However, we note that while the features we examined are wellstudied hallmarks of degeneration, they are specific rather than exhaustive, and subtle morphological characteristics that are beyond our methods’ detection may change. We have added these new results to Figure 2C and added these notes to the manuscript.

The histologic analysis is based on mice with different genetic backgrounds. For example, the PC-specific reporter mice include two strains: Pcp2-Cre; Ai32 and Pcp2-Cre; Ai40D. These genetic variations may contribute to the heterogeneity of PC loss (Figure 1). To improve clarity, please add the genetic background details to Table 1.

We have added the genetic backgrounds of all mice used in the study to Table 1.

Please indicate from which lobule in the anterior or posterior human cerebellum the images in Figure 8 were taken.

Unfortunately, because of the limitations of human postmortem tissue collection (in some cases, we are provided with a very small block that was collected after the pathologist completed their primary duty for that individual), we cannot with full certainty distinguish the lobules from which the images were taken. However, we are grateful that, upon our request, the pathologists were able to collect tissue mainly from the vermis, which is where we wished to begin, knowing that the vermis in rodents and non-human primates typically has the clearest and most well-studied pattern. That said, this is an important issue that we are addressing for future studies.

**Reviewer #2 (Public review):**
(1) Limited strain diversity: The study exclusively uses C57BL/6J mice despite known genetic and motor differences even the closely related strains like C57BL/6N.

Thank you for pointing out this limitation of our study. We chose to limit this initial study to C57BL/6J mice based on their widespread use as a background strain on many currently maintained lines. That said, our study intentionally included several different crosses to provide genetic variability, even though C57BL/6J is still the predominant genetic background. In addition to the motor differences in genetic strains, we are also particularly interested in the differences in cerebellar morphology across strains (Inouye and Oda, 1980; Sillitoe and Joyner, 2007). Our use of mice maintained on the C57BL/6J background leaves open an exciting future direction: investigating age-related Purkinje cell loss in mice of different inbred and outbred strains. Given the importance of the topic, we have included new text in the discussion to alert the reader to this limitation of our study and to highlight interesting differences across strains that will be important to disentangle in our future work.

(2) No correlation quantified between the degree of PC loss, aging, and motor performance.

We sought to conduct a broad overview of motor problems that might be caused by age-related Purkinje cell loss, rather than a comprehensive investigation of how motor behavior changes with advancing Purkinje cell loss. Therefore, we agree with the reviewer’s comment, and we have added text to indicate that stronger correlations between these domains would be best tackled with deeper behavioral phenotyping conducted over time to match the potentially cooccurring progressive changes in cerebellar morphology, with a focus on Purkinje cell degeneration and eventual loss.

(3) It has not been demonstrated whether the neurodegenerative changes are indeed observed in zebrin-negative PCs.

We have added Supplementary Figure 4, which includes an example of reduced dendritic density and loss of Purkinje cell somata in zebrin II-negative stripes in lobules II and III. Please also see Figure 4B for an example of reduced dendritic density in zebrin II-negative Purkinje cells in lobules III and IV.

(4) The mechanisms of why only a subset of mice show PC loss remain unexplored and not discussed.

We agree that our manuscript would benefit from discussion of why some aged mice are resistant to age-related Purkinje cell loss. We have elaborated upon possible reasons for this differential vulnerability in the discussion.

(5) Linkages with normal human aging and cerebellar function are not well supported. While motor behavioral assays captured phenotypes that mimic aged people, correlation with PC loss is demonstrated to be absent in mice. It remains unclear whether this PC loss phenomenon is universal or specific to a particular individual; and whether specific to a human PC subtype.

In our study, we sought to show that patterned age-related Purkinje cell loss presents a promising area for future research in humans. We agree that further study is needed to solidify a link between age-related Purkinje cell loss in mice and humans and the implications for motor function. The reviewer raises a fair criticism that reflects the current state of knowledge: studies that link cerebellar aging to motor function and cognitive decline in humans are few, as are studies of the cellular-level morphological changes of cerebellar aging –there is a pressing need for deeper study of human tissue. To address the issue raised by the reviewer, we have included new text to the discussion of our manuscript indicating these gaps in knowledge.

(6) Analyses in the paraflocculus are currently not easy to understand. This lobule has heterogeneous PC subtypes, developmentally or molecularly. Zebrin-weak and Zebrinintense PCs are known to be arranged in stripes, which resembles the pattern of developmentally defined PC subsets (Fujita et al., 2014, Plos one; Fujita et al., 2012, J Neurosci). In the data presented, it is hard to appreciate whether the viewing angle is consistent relative to the angle of the paraflocculus. This may be a limitation of the analysis of the paraflocculus in general, that the orientation of this lobule is so susceptible to fixation and dissection. Discrepancy between PC loss stripe and zebrin pattern may be an overstatement, because appropriate analyses on the paraflocculus would require a rigorously standardized analytic method.

Thank you for your valuable insights on the complexity of analyzing the paraflocculus. We have altered our language to more accurately reflect the nuanced zebrin II expression pattern of this region. We also agree with and very much appreciate your advice that “analyses on the paraflocculus would require a rigorously standardized analytic method.” We have added these arguments to the revised manuscript text.

**Reviewer #3 (Public review):**
(1) In Figure 3, the authors use Pcp2-CRE mice to drive GFP expression in Purkinje cells in order to avoid the confounding variable of loss of calbindin expression in aging Purkinje cells. The authors go on to say, "we argue that calbindin expression alone is not a reliable, sufficient indicator of Purkinje cell loss". However, in Figure 4, the authors return to calbindin staining alone to assess the correlation of Purkinje cell loss with zebrin-II expression. Could the authors comment on why zebrin-II co-staining experiments were not performed in GFP reporter mice to avoid potential confounds of calbindin expression? Without this experiment, should readers accept the data presented in Figure 4 as a "reliable, sufficient indicator of Purkinje cell loss", given the author's prior claim?

This is a very good point, thank you. We agree that the data presented in Figure 4 alone would not be a sufficient indicator of Purkinje cell loss. However, we prefaced our calbindin and zebrin II co-staining with calbindin and GFP costaining (Figure 3), which showed that Purkinje cell-specific reporter expression revealed the same pattern of Purkinje cell loss as calbindin expression, and Neutral Red staining (Figure 2 and Supplementary Figure 3B), which confirmed the loss of Purkinje cells independent of immunofluorescence. For this reason, we feel confident that the data in Figure 4 is representative of the striped pattern of age-related Purkinje cell loss. Still, we see and agree with the reviewer’s comment, and therefore, to further show the correlation of Purkinje cell loss with zebrin II expression, we have added a new Supplementary Fig. 4, which shows co-staining of calbindin, GFP, and zebrin II.

(2) Throughout the manuscript, there is a considerable reliance on the authors' interpretation of imaging data with no accompanying quantification (categorization of "striped" or "non-striped" PC loss, correlation of GFP/calbindin/zebrin-II staining, etc.). While this may be difficult to obtain, the results would be much stronger with a quantitative approach to support the stated categorizations/observations.

Thank you for your suggestion. Quantifying stripe properties has been a challenging task for the field, given the regionalized features of stripe compartmentalization that make its complex architecture tricky to measure in its typical organization within the 3D anatomy of lobules and fissures and even harder to interpret when there are abnormalities. However, to quantitatively support our categorization of “striped” and “non-striped” Purkinje loss and the observed correlation between calbindin and GFP expression in aged mice, we have quantified the mediolateral pixel intensity across lobules II-IV, in which Purkinje cell loss reliably occurs in zebrin II-negative stripes. The results can be found in Supplementary Figure 1B and Supplementary Figure 3.

**Reviewer #1 (Recommendations for the authors):**
(1) In Figure 1, both staining artifacts and PC degeneration appear in light color. Please clarify how these two were differentiated.

Thank you for your comment, which raises an important point about distinguishing staining artifacts from Purkinje cell degeneration. Cerebellar patterning is symmetrical across the midline, so asymmetrical abnormalities are one clue that differentiates staining artifacts from the degenerative pattern. Another indicator of a staining artifact seen in wholemount preparations is the gradual fading of the stain (seen in some hemispheres in Figure 1), which is caused by continuous rubbing of the cerebellum against the tube during the staining process. In some cases, such as in Figure 1F, the cerebellum was damaged during the dissection of the meninges after staining, and in such cases the accidental removal of cerebellar tissue (molecular layer) reveals unstained tissue beneath the surface of the cerebellum. This type of staining artifact can be identified by a missing chunk of tissue surrounded by stained Purkinje cells, compared to the smooth, unmarred tissue where PCs have degenerated. We have added new text to the results (the legends) to clarify these critical points for the reader.

(2) In Figure 7C, please consider changing "Aged without stripes" to "Aged without PC loss" to be consistent with the labeling used in other panels.

Thank you for pointing out this discrepancy. We have made the suggested changes.

**Reviewer #3 (Recommendations for the authors):**
Could the authors comment on why zebrin-II co-staining experiments were not performed in GFP reporter mice to avoid potential confounds of calbindin expression? Without this experiment, should readers accept the data presented in Figure 4 as a "reliable, sufficient indicator of Purkinje cell loss", given the author's prior claim?

Thank you for this recommendation; we appreciate this advice. As we described above, our response to this suggestion reads:

This is a very good point, thank you. We agree that the data presented in Figure 4 alone would not be a sufficient indicator of Purkinje cell loss. However, we prefaced our calbindin and zebrin II co-staining with calbindin and GFP costaining (Figure 3), which showed that Purkinje cell-specific reporter expression revealed the same pattern of Purkinje cell loss as calbindin expression, and Neutral Red staining (Figure 2 and Supplementary Figure 3B), which confirmed the loss of Purkinje cells independent of immunofluorescence. For this reason, we feel confident that the data in Figure 4 is representative of the striped pattern of age-related Purkinje cell loss. Still, we see and agree with the reviewer’s comment, and therefore to further show the correlation of Purkinje cell loss with zebrin II expression, we have added a new Supplementary Fig. 4, which shows co-staining of calbindin, GFP, and zebrin II.